# Lagrangian formation pathways of moist anomalies in the trade-wind region during the dry season: two case studies from EUREC⁴A

Leonie Villiger[1], Heini Wernli[1], Maxi Boettcher[1], Martin Hagen[2], and Franziska Aemisegger[1]

[1]Institute for Atmospheric and Climate Science, ETH Zurich, Zurich, Switzerland
[2]Institut für Physik der Atmosphäre, Deutsches Zentrum für Luft- und Raumfahrt, Oberpfaffenhofen, Germany

**Correspondence:** Leonie Villiger (leonie.villiger@env.ethz.ch)

**Abstract.** Shallow clouds in the trade-wind region over the North Atlantic contribute substantially to the global radiative budget. In the vicinity of the Caribbean island Barbados, they appear in different mesoscale organisation patterns with distinct net cloud radiative effects (CRE). Cloud formation processes in this region are typically controlled by the prevailing large-scale subsidence. However, occasionally weather systems from remote origin cause significant disturbances. This study investigates the complex cloud-circulation interactions during the field campaign EUREC⁴A (Elucidate the Couplings Between Clouds, Convection and Circulation) from 16 January to 20 February 2020, using a combination of Eulerian and Lagrangian diagnostics. Based on observations and ERA5 reanalyses, we identify the relevant processes and characterise the formation pathways of two moist anomalies above the Barbados Cloud Observatory (BCO), one in the lower ($\sim$1000-650 hPa) and one in the middle troposphere ($\sim$650-300 hPa). These moist anomalies are associated with strongly negative CRE values and with contrasting long-range transport processes from the extratropics and the tropics, respectively. The first case study about the low-level moist anomaly is characterised by an unusually thick cloud layer, high precipitation totals and a strongly negative CRE. The formation of the low-level moist anomaly is connected to an "extratropical dry intrusion" (EDI) that interacts with a trailing cold front. A quasi-climatological (2010-2020) analysis reveals that EDIs lead to different conditions at the BCO depending on how they interact with the associated trailing cold front. Based on this climatology, we discuss the relevance of the strong large-scale forcing by EDIs for the low-cloud patterns near the BCO and the related CRE. The second case study about the mid-tropospheric moist anomaly is associated with an extended and persistent mixed-phase shelf cloud and the lowest daily CRE value observed during the campaign. The formation of the mid-level moist anomaly is linked to "tropical mid-level detrainment" (TMD), which refers to detrainment from tropical deep convection near the melting layer. The quasi-climatological analysis shows that TMDs consistently lead to mid-tropospheric moist anomalies over the BCO and that the detrainment height controls the magnitude of the anomaly. However, no systematic relationship was found between the amplitude of this mid-tropospheric moist anomaly and the CRE at the BCO. This is most likely due to the modulation of the CRE by above and below lying clouds and the fact that we used daily mean CRE thereby ignoring the impact of the timing of the synoptic anomaly with respect to the daily cycle. Overall, this study reveals the important impact of the long-range moisture transport, driven by dynamical processes either in the extratropics or the tropics, on the variability of the vertical structure of moisture and clouds, and on the resulting CRE in the North Atlantic winter trades.

# 1 Introduction

The representation of trade-wind cumuli in model simulations critically influences climate projections (e.g., Bony et al., 2015; Schneider et al., 2017; Zelinka et al., 2017). These clouds form from the complex interplay of turbulent mixing, shallow convection, cloud radiative processes and large-scale subsidence within the descending branch of the Hadley circulation (Webb et al., 2015; Vial et al., 2016, 2017). In return, they modulate the thermodynamic conditions in their environment (Bony et al., 2017) and can induce mesoscale circulations (Naumann et al., 2017) through diabatic processes. Thus, clouds and the atmospheric circulation are closely tied to each other on various scales.

This study investigates the influence of meridional long-range transport on the formation of moist anomalies and trade-wind clouds during the field campaign EUREC[4]A (Elucidate the Couplings Between Clouds, Convection and Circulation; Bony et al., 2017; Stevens et al., 2021), which took place in early 2020 in the vicinity of the Caribbean island Barbados. A particular focus is set on periods when the thermodynamical vertical profile deviates from what is typically expected in the trade-wind region, i.e., a shallow cloud layer with a dry free troposphere aloft. The study has three research aims: (1) to characterize the variability in the large-scale circulation during EUREC[4]A and its influence on the variability in the thermodynamical vertical profile over Barbados; (2) to analyse the transport and processes during this transport that lead to the formation of moist anomalies and clouds in the lower and middle troposphere, respectively; and (3) to quantify the climatological frequency of the identified transport pathways and of the robustness of their link with moist anomalies. In the following paragraphs, we provide further information about the field campaign as well as the processes related to dry intrusions and detrainment from deep convective clouds near the melting layer, which are both essential features for the moist anomalies analysed in this study.

Providing simultaneous observations from multiple platforms complemented with a substantial modelling effort, EUREC[4]A aims at deepening our understanding of the cloud-circulation coupling in the trades (Bony et al., 2017; Stevens et al., 2021). Already before the field campaign, the clouds near Barbados were a matter of active research. It is known that they organise in four different mesoscale patterns (Stevens et al., 2020) with distinct radiative effects (Bony et al., 2020) and in response to differing environmental conditions (Schulz et al., 2021). Furthermore, the cloudiness in this region is known to undergo a daily cycle, which peaks in the early morning (Vial et al., 2019). Barbados represents an ideal site for studying shallow trade-wind cumuli (Medeiros and Nuijens, 2016; Stevens et al., 2016) and has been the base for many previous field experiments, e.g., BOMEX (1969; Holland, 1970), RICO (2004/2005; Rauber et al., 2007), CARRIBA (2010/2011; Siebert et al., 2013), Narval I (2013; Stevens et al., 2016) and II (2016; Stevens et al., 2019).

Due to its location at the edge of the tropics, the region of Barbados experiences large-scale subsidence at upper levels and prevailing low-level easterlies during boreal winter. The subsidence rate at upper levels results from the balance between radiative cooling and adiabatic warming and is estimated to be $35\,\mathrm{hPa}\,(24\,\mathrm{h})^{-1}$ (Salathé and Hartmann, 1997). However, this typical trade-wind regime is frequently disturbed by intrusions form either sides, i.e., the extratropics or deep tropics. The Lagrangian perspective is useful to investigate such disturbances as it allows us to identify the origin and transport history of individual air parcels eventually arriving in the trade-wind region. This approach has been adopted previously in several studies, e.g., to investigate the origin of dry and humid layers in the (sub)tropics (Yoneyama and Parsons, 1999; Waugh, 2005; Cau

et al., 2005, 2007; Casey et al., 2009) or to link the cloudiness in Barbados to the large-scale transport (e.g., Aemisegger et al., 2021; Schulz et al., 2021). Here, we use it to study the formation of moist anomalies above Barbados and, more specifically, to identify air parcels that (1) rapidly descend from the upper-level extratropics towards Barbados, so-called dry intrusions (DI, e.g., Browning and Roberts, 1994; Browning and Golding, 1995; Wernli, 1997, we use here the abbreviation EDI to emphasize the extratropical origin); or (2) flow out of deep and mid-level convective systems over tropical South America and spread over Barbados at mid levels (e.g., Johnson et al., 1996, 1999; Stevens et al., 2017). We refer to this phenomenon as tropical mid-level detrainment (TMD).

EDIs are coherent dry airstreams that descend from near-tropopause levels into the middle or lower troposphere in the rear of extratropical cyclones (e.g., Browning, 1997; Raveh-Rubin, 2017). They are visible in water vapour satellite images as dry slots with a moisture front at their leading edge (Browning and Golding, 1995; Browning, 1997). The dry layer induced by the EDI potentially increases outgoing longwave radiation up to $3\,\mathrm{W\,m^{-2}}$ per $100\,\mathrm{m}$ of the layer's depth (Cau et al., 2005). As shown by Browning and Golding (1995), EDI air parcels have a distinct impact on surface precipitation depending on whether they under- or overrun the so-called warm conveyor belt (WCB), i.e., a moist ascending airstream ahead of the cold front. In the first case, the EDI might suppress precipitation by evaporating the falling hydrometeors. In the second case, potential instability builds up that results in convective precipitation if it is released. At times, EDIs are related to upper-level Rossby-wave breaking, which forms narrow tongues of stratospheric air with high potential vorticity (PV). If such a tongue of high PV air, also referred to as a PV streamer (Appenzeller and Davies, 1992) extends into the tropics, it can trigger tropical convection (e.g., Kiladis and Weickmann, 1992; Waugh and Polvani, 2000).

In the Lagrangian framework, EDIs have been identified as air parcels descending $400\,\mathrm{hPa}$ within two days ($\Delta p \geq 400\,\mathrm{hPa}$ $(48\,\mathrm{h})^{-1}$, following Raveh-Rubin, 2017). Using this method, Raveh-Rubin (2017) provided a climatology showing that EDIs are most frequent during boreal winter. They descend nearly adiabatically at 40-50° N following the slanting isentropes towards lower latitudes, experience a decrease in relative humidity on the first day, followed by a moistening due to mixing with planetary boundary layer (PBL) air on the second day, and typically induce low-level instability, wind gusts, intense upward surface latent and sensible heat fluxes, and a deepening of the PBL [recently also found in a study combining observational and reanalysis data by Ilotoviz et al. (2021)]. Distinguishing between trailing cold fronts (connected to an extratropical cyclone) and isolated cold fronts (no cyclone nearby), Catto and Raveh-Rubin (2019) and Raveh-Rubin and Catto (2019) found that trailing fronts match more frequently with EDIs than isolated fronts and suggested that EDIs are essential for the appearance of trailing fronts in the subtropics. Both front types are elongated and strengthened through the deformation flow imposed by the EDI (Catto and Raveh-Rubin, 2019; Raveh-Rubin and Catto, 2019). This strengthening of the fronts, i.e., frontogenesis, can be understood following quasi-geostrophic theory (Davies and Wernli, 2015), if two necessary ingredients are given: a horizontal temperature gradient in combination with a suitably oriented deformation flow. Frontogenesis is associated with an ageostropic circulation inducing low-level convergence, ascending motion and potentially cloud formation on the front's warm side. The role of convergence lines, in a general sense, has been studied for precipitation in the subtropics (Weller et al., 2017). In our first case study, we illustrate that some of these convergence lines are linked to EDI-related frontogenesis.

Some careful considerations are needed before using the term cold front for features in the subtropics. Commonly cold fronts are identified as regions in the extratropics with strong horizontal gradients in low-level equivalent potential temperature (see, e.g., Schemm et al., 2015). However, strong horizontal gradients in equivalent potential temperature and low-level convergence lines, comparable to the ones in the extratropics, are found in the subtropics too. These features may be interpreted as (1) the extension of an extratropical cold front into the subtropics (the so-called trailing cold front), (2) the southern edge of the cold sector of an extratropical cold front, (3) the EDI spreading out behind the extratropical cold front, or (4) the boundary between extra- and subtropical air masses pushed southwards under the impact of the EDI. In the following we use the term (trailing) cold front, but recall that their dynamical properties may differ from extratropical cold fronts.

Still, concepts developed for extratropical fronts might be useful to characterize trailing cold fronts in the subtropics and their interaction with the associated EDI, for example, the differentiation between ana- and katafronts (Bergeron, 1937; Sansom, 1951) or surface and upper-level fronts (Browning and Monk, 1982; Browning and Roberts, 1996). Anafronts are characterized by warm air ascending along the backward tilted cold front, resulting in a narrow rain band along the surface cold front and stratiform precipitation behind it (Sansom, 1951; Browning and Roberts, 1996), possibly leading to an evaporative cooling of the cold sector and a sharpening of the horizontal temperature gradient. At katafronts, in contrast, the warm air's ascent is vertically limited due to the descending EDI which overruns the surface cold front. This process leads to a destabilisation of the shallow moist zone ahead of the surface cold front and the formation of a second front aloft represented by the EDI's leading edge. This situation is also referred to as a split-front (Browning and Monk, 1982, their Fig. 9). According to Browning and Roberts (1996), precipitation associated with katafronts can be stratified into three regimes: a narrow precipitation band along the surface cold front, individual convective cells in the destabilized shallow moist zone and deeper convection at the cold front aloft. Locatelli et al. (1997) provided the theoretical background for the occurrence of low-level convergence ahead of an upper cold front overrunning warm air, which nicely fits the convection and moisture accumulation at the leading edge of the EDI. As mentioned before, these concepts were defined based on observations of extratropical cold fronts. Whether trailing cold fronts in the subtropics adopt similar characteristics as the two extratropical front types is discussed in Sect. 4.3.

After having introduced the EDI, a flow feature from the extratropics, we now turn our attention to tropical convection and how its outflows can affect the trade-wind region in the form of TMDs. The vertical distribution of tropical clouds is known to be tri-modal (Johnson et al., 1999) with deep cumulonimbi reaching up to the tropopause, shallow cumuli being limited in their vertical growth by the trade inversion (up to max 700 hPa, Schubert et al., 1995), and a third maximum in cloudiness emerging near the melting layer (typically around 500-600 hPa, Zuidema, 1998). This layer is often characterized by a stable stratification, capping cumuli congesti and promoting detrainment from cumulonimbi (resulting in so-called shelf-clouds, Johnson et al., 1999). Initial stabilising processes are linked to diabatic warming related to falling ice crystals, which trigger freezing (through contact with supercooled water) above the $0°$ C isotherm and diabatic cooling due to melting below it (Johnson et al., 1996; Posselt et al., 2008; Stevens et al., 2017). The presence of a mid-tropospheric dry layer, into which the ice crystals are falling, contributes to the efficiency of the sublimation cooling (Zuidema et al., 2006). Once the stable layer is established, outflow from cumulonimbi at this level is promoted and the detrained cloud masses exert an overhead radiative cooling, which maintains stability and initiates subsidence. The resulting overturning circulation enhances TMD due

to horizontal divergence near the melting layer (Posselt et al., 2008; Stevens et al., 2017). In early studies, additional processes that possibly help to maintain and horizontally expand the stable layer near the melting layer away from the deep convective precipitating system, were mentioned, e.g., the gravity-wave like propagation of a single heating event in a stratified fluid (Mapes, 1993; Mapes and Houze Jr., 1995). In our second case study, we demonstrate how TMD near the melting layer affects the moisture budget of the lower free troposphere in the trade-wind region.

This paper continues with a description of the data and methods (Sect. 2). After an overview of the humidity profiles and transport conditions during the campaign (Sect. 3), the first part of the results focuses on the effect of EDIs on the lower-tropospheric humidity (Sect. 4). The second part looks into the effect TMDs have on the mid-tropospheric humidity over Barbados (Sect. 5). Both parts include a detailed, illustrative case study from EUREC⁴A, which is complemented with a quasi-climatological analysis. The paper ends with a summary and concluding remarks (Sect. 6).

## 2 Data and methods

We use the hourly ERA5 reanalysis data set (Hersbach et al., 2020) on a regular $0.5° \times 0.5°$-grid to characterise the atmospheric column above Barbados and to compute three-dimensional air parcel backward trajectories from Barbados. For comparison, we consult observational data sets from two sites of the EUREC$^A$A experiment and different satellite products. The BCO (https://barbados.mpimet.mpg.de/), situated at the east coast of Barbados (13.16° N, 59.43° W, 17 m a.s.l.; Stevens et al., 2016), is operated by the Max Planck Institute for Meteorology, the Caribbean Institute for Meteorology and Hydrology and the Museum of Barbados. From this site, we use the 10 s precipitation measurements taken by the Vaisala WXT-520 meteorological ground station (4 m. a.g.l., Jansen, 2020), the vertical profiles from a Ka-band cloud radar MIRA-36 (11 m. a.g.l.; METEK GmbH; www.metek.de; hereafter referred to as cloud radar; Stevens et al., 2016), and the data obtained by the Vaisala RS41 radiosondes launched from the BCO (Stephan et al., 2020, typically six radiosondes within 24 h). Data from the Polarization Diversity Doppler Radar (Poldirad; Hagen et al., 2021), deployed by the German Aerospace Center (Deutsches Zentrum für Luft- und Raumfahrt) roughly 8 km west-northwest of the BCO (13.18° N, 59.5° W, 245 m a.s.l.), expand our analysis beyond the location of the BCO. The variables CRE and total ice water are taken from the satellite product Clouds and the Earth's Radiant Energy System (CERES; NASA, 2017) distributed by the National Aeronautics and Space Administration (NASA). CERES data is available at an hourly temporal and a $1° \times 1°$ spatial resolution. Finally, images from the MODIS instrument on the satellite Terra provided by the Earth Observing System Data and Information System (EOSDIS; https://wvs.earthdata.nasa.gov) and from the GOES-16 satellite provided by the GIBBS imagery service (https://www.ncdc.noaa.gov/gibbs/; Knapp, 2008) give an overview of the cloudiness on the synoptic scale. In the following, we describe how the local conditions and the transport pathways towards Barbados were characterized.

### 2.1 Characterisation of the local conditions in Barbados

The ERA5 variables are tri-linearly interpolated to a vertical profile (1000-300 hPa, every 7 hPa; Fig. 1a) at the geographical location of the BCO every hour in the EUREC⁴A period (16 January to 20 February 2020), and every third hour in the quasi-

climatological period (January and February 2010 to 2020). For the case study analyses (Sect. 4 and 5), we average or sum the variables over each day and two vertical layers, namely a lower-tropospheric (1000-650 hPa) and a mid-tropospheric (650-300 hPa) layer. The 24 h time window is chosen because the associated large-scale conditions typically remain similar within one day (Fig. 4). Moreover, the cloud radiative effect should be assessed combining day- and night-time conditions. The two vertical layers are defined such that both contain 50 vertical data points and are motivated by the fact that the maximal vertical extent of the low-level cloud layer during the campaign is at about 650 hPa (Fig. 2). This simple definition of temporal and vertical boundaries to obtain a summarizing statistics is also applied at the interannual timescale to investigate the robustness of the identified links between the large-scale circulation and the local conditions. We repeated the above-described steps for four additional points shifted zonally and meridionally from the BCO by 0.5° and found that the results for the five locations differ only marginally (not shown). We therefore decide to focus in this study on the profiles directly above the BCO only. The following variables are computed with ERA5 data and partly also with measurements.

**Integrated water vapour (IWV):** We calculate the IWV for the two layers ($IWV_{1000\text{-}650\,hPa}$, $IWV_{650\text{-}300\,hPa}$) using the equation $IWV_{p_{bottom}-p_{top}}[m] = \frac{1}{g\rho_l}\int_{p_{bottom}}^{p_{top}} q\,dp$, with specific humidity ($q$), density of liquid water ($\rho_l$), pressure ($p$), and the gravitational acceleration ($g$). We derive the IWV from ERA5 and the radiosonde measurements (Stephan et al., 2020, 2021). Daily mean IWV values are used in both case studies (Sect. 4 and 5).

**Total ice (liquid) water:** For ERA5, total ice (liquid) water is calculated by adding total column cloud ice water and the vertically integrated specific snow (rain) water content, which is obtained following the procedure for IWV. ERA5's total ice (liquid) water is compared to CERES' ice (liquid) water path, in the following also referred to as total ice (liquid) water. Total ice water is used in the TMD case study (Sect. 5 and in Supplement 1 together with total liquid water).

**Precipitation totals:** ERA5 daily precipitation totals are obtained from adding up convective and large-scale precipitation. Daily precipitation totals are also retrieved from Vaisala WXT-520 measurements (Jansen, 2020) and are used in the overview of the EUREC[4]A period (Sect. 3) and the EDI case study (Sect. 4).

**Net cloud radiative effect (CRE):** Following Boucher et al. (2013) and Hartmann (2016), we calculate the net cloud radiative effect (CRE) as the sum of the longwave (LWCRE) and shortwave (SWCRE) cloud radiative effect. The LWCRE results from the difference of the top of atmosphere thermal radiation under clear-sky conditions and the one under cloudy conditions (clear-sky minus cloudy), and analogously for the SWCRE and solar radiation. Using this convention, negative values represent a radiative cooling due to the presence of clouds and positive values a warming. As the cloud radiative forcing is very inhomogeneous in space (see Fig. S1.7-S1.8), an area-weighted mean over the domain 10-20° N, 50-60° W is calculated. The domain's size is chosen similar to Bony et al. (2020), who used 10-20° N, 48-58° W, but here the borders are shifted to the west to include Barbados. The domain-mean CRE allows a comparison with the results from Bony et al. (2020) and is used in both case studies (Sect. 4 and 5). The CRE was also derived from the satellite product CERES (NASA, 2017).

## 2.2 Characterisation of the transport pathways towards Barbados

Three-dimensional backward trajectories are computed with the Lagrangian analysis tool LAGRANTO (Wernli and Davies, 1997; Sprenger and Wernli, 2015), starting from the interpolation points defined above (Fig. 1b). The calculations are based on

the ERA5 wind fields and determine the air parcels' position (longitude, latitude, pressure) during the ten days before arrival. A set of variables is interpolated along the trajectories, namely, specific humidity (absolute values and anomalies relative to the three-dimensional campaign mean field), relative humidity, liquid/rain/snow/ice water content, and surface evaporation. We extract each air parcel's position four days before arrival (Fig. 4; see Fig. S1.1 for other time steps) to get an impression of the pathway variability across the vertical profile and during the EUREC⁴A time period. The choice of four days is inspired

by the study of Aemisegger et al. (2021), who showed that the transport during this pre-arrival time window is essential for understanding moisture anomalies induced by the large-scale circulation in the trades. Typically, multiple airstreams (coherent bundles of trajectories with similar characteristics) arrive in each of the two layers during 24 h. For each airstream, defined in Sect. 4.2 and 5.2, we compute the mean specific humidity (per air parcel) and the contribution to the IWV of the arrival layer (Table 1 and 4).

## 2.3 Representativeness of ERA5 in the vicinity of Barbados

The findings of this study are mainly based on ERA5 reanalyses, therefore some words about their validity are needed. Several comparisons between ERA5 and measurements of the local conditions in Barbados, made in the context of the two case studies (e.g., Fig. 3, 6, 13, and Fig. S1.3-S1.8), show that ERA5 reproduces the variability fairly well (including the timing of local extremes), but differs in terms of absolute values. ERA5 underestimates IWV, especially in the lower troposphere, and

210 has difficulties to capture the extremely high values of specific humidity observed by the BCO radiosondes (Stephan et al., 2021), e.g. from 22 to 24 January 2020, in both layers (Fig. 3a,b). This lower-tropospheric dry bias of ERA5 has been noticed previously (Bock et al., 2021; Chellappan et al., 2021). Besides this systematic bias, the otherwise good agreement of IWV variability does not come as a surprise, as the soundings from the nearby Grantley Adams International Airport of Barbados (but not from the BCO) are assimilated in ERA5 (Bock et al., 2021).

Precipitation totals (Fig. 3c, 6b) are underestimated by ERA5 compared to the in-situ measurements of the meteorological station WXT-520, which is most probably an effect of ERA5's spatial and temporal resolution. While the reanalysis yields an average over a model grid box at every hour, the WXT-520 provides a point measurement every 10 s. A direct comparison is therefore difficult. Similarly, ERA5 produces too low total ice water values (Fig. 3d, 13b) with CERES as a reference. Note, however, that most of the time CERES total ice water information in the vicinity of Barbados is missing and only spatially

interpolated data is available (red dots in Fig. 3d). Thus, the explanatory power of CERES total ice water shown in Fig. 3 is very limited and it is hard to judge, which of the two data sets is closer to real conditions.

On the hourly time scale, ERA5's underestimation of atmospheric ice in mid- to high-level clouds (which potentially reduce the cooling effect of low-level clouds, see Adebiyi et al., 2020) results in an underestimation of the cloud radiative warming during local night-time (see, e.g., 14 February in Fig. 3d, Supplement 1) compared to CERES. During local daytime, ERA5

often yields a stronger cloud radiative cooling, which might be the result of too high low-level cloud fractions (as observed, at least, in one case, see Fig. S1.7). Eventually, these differences lead to deviating daily mean CRE for ERA5 and CERES (Fig. 6, Fig. 13, Supplement 1). Important to note, however, is that time steps with strong anomalies in ERA5 are also identified as anomalies in CERES, e.g. on 22 January and 14 February 2020. Moreover, based on the observed differences in absolute

values, we assume that the anomalies of IWV, precipitation, total ice water and (on 22 January) the cloud radiative cooling, discussed in the two case studies, were stronger in reality than indicated by ERA5.

Even harder to evaluate is the accuracy of the backward trajectories calculated with ERA5 wind fields. As a first step, we compared the horizontal ERA5 wind components to the ones measured by the BCO radiosondes (Fig. S1.4), which showed that the two data sets agree in terms of variability and absolute values. To assess the vertical velocities, we compared the mean pressure velocities of the trajectories during the last 24 h before arrival (i.e., the period when the trajectories are already in the vicinity of Barbados) to measurements (valid over a circular area with a diameter of $\sim 175\,\text{km}$; George et al., 2021) from previous field campaigns (Fig. S1.5). The pressure velocities derived from the trajectories are mostly within the range of roughly $-5$ to $5\,\text{hPa}\,\text{h}^{-1}$ determined by George et al. (2021, their Fig. 3) using dropsonde measurements. Moreover, we know from previous studies that ERA5 agrees reasonably well with dropsonde-derived divergence measurements, which directly link to vertical motion (Li et al., 2021). Thus, we are confident that the ERA5 wind fields near Barbados are close to the real conditions and therefore lead to representative trajectories. The informative value of ERA5 trajectories was recently also demonstrated by Hoffmann et al. (2019), who showed that ERA5 trajectories are physically more consistent (e.g., improved potential temperature conservation along trajectories in the stratosphere) than ERA-Interim trajectories, most probably due to a higher temporal resolution of the wind fields. Finally, Aemisegger et al. (2021) could explain measured variations of stable water isotope signals in Barbados with ERA5 backward trajectories, which supports the usefulness of the trajectories.

## 3 Temporal evolution of the atmospheric column above Barbados during EUREC[4]A

In this section, we present the humidity, wind, radiative and transport conditions over Barbados during EUREC[4]A. Throughout the campaign, the Eulerian (Fig. 2 and 3) and the Lagrangian (Fig. 4) conditions varied substantially in the lower and middle troposphere. The clouds over the BCO can be identified from contours of liquid water content and were generally associated with localised positive anomalies in specific humidity (Fig. 2a). The cloud tops were typically below $700\,\text{hPa}$ but occasionally reached $600\,\text{hPa}$, namely on 22-24 January, 11-13 February and 18-19 February. The extended cloud layers led to elevated values of $\text{IWV}_{1000\text{-}650\,\text{hPa}}$ (Fig. 3a) and precipitation totals (Fig. 3b), whereof the most persistent rain event took place on 22 January. The daily mean cloud radiative cooling (Fig. 3c) was comparably strong during these three periods (about $-29$ to $-45\,\text{W}\,\text{m}^{-2}$ compared to a campaign-mean value of $-21\,\text{W}\,\text{m}^{-2}$ according to ERA5).

Towards the end of the campaign, a persistent mixed-phase cloud layer (Fig. 2a) appeared at mid-tropospheric levels leading to a positive anomaly in $\text{IWV}_{650\text{-}300\text{hPa}}$ (Fig. 3a) with dry conditions below. This mid-tropospheric cloud provoked an immediate response in the CRE (Fig. 3c) with stronger warming/cooling during night-/daytime (local time in Barbados corresponds to UTC-4 h), leading to a daily mean CRE of $-79\,\text{W}\,\text{m}^{-2}$ in ERA5 and $-61\,\text{W}\,\text{m}^{-2}$ in CERES on 14 February, the most negative value during the campaign.

The local wind conditions indicate periods with different strength of the prevailing low-level easterlies (Fig. 2b). From 25 January to 4 February the low-level easterlies were weak, followed by a period with strong easterlies until 20 February. At mid-tropospheric levels, northwesterlies dominated, except for 12-16 February, when the mixed-phase cloud layer advected from the

south was present. Overall, weak downward vertical winds were observed (Fig. 2c). At low levels, upward motion associated with convective cells and rain water content, e.g., on 22 January, alternated with downward motion in downdrafts. On 12-16 February, unusually pronounced upward winds above approximately $550\,\text{hPa}$ occurred together with enhanced subsidence below.

The distance the air parcels traveled within the four days before arrival (Fig. 4a) shows similar variability patterns as the local horizontal winds (Fig. 2b and c), including periods with close to stationary transport conditions within the last days prior to arrival and periods with pronounced long-range transport exceeding $5000\,\text{km}$ in four days. At upper levels, the air parcels generally traveled long distances, except when they were advected from the tropics (12-16 February, Fig. 4b). At low levels, weak local easterlies (Fig. 2b) coincided with short-range transport and strong easterlies with long-range transport (Fig. 4a). One exception can be found around 22 January, when long-range transport is observed at low levels, even though the local easterlies are not particularly strong. These air parcels originated from high latitudes (Fig. 4b) and descended rapidly (Fig. 4c) until reaching the BCO. On 8 February, another (less coherent) descending airstream from the extratropics arrived in Barbados. Otherwise, the air parcels moved within the subtropics or tropics (Fig. 4b) and subsided at a moderate rate corresponding to the expected values (of roughly $35\,\text{hPa}\,(24\,\text{h})^{-1}$, see Salathé and Hartmann, 1997; Holton and Hakim, 2013) from the balance between adiabatic compression and radiative cooling (Fig. 4c).

Based on the descriptions above, the typical trade-wind conditions (26 January to 7 February) with slow descending motion and weak horizontal transport in the lower free troposphere combined with low-level easterlies (Hadley-like circulation) are associated with a relatively moist well-mixed sub-cloud layer, a shallow cloud layer (950-750 hPa), and a strong inversion associated with very dry conditions (at 600 to 900 hPa; Fig. 2 and 4). These typical trade wind conditions are interrupted during two periods (indicated by the red rectangles or shading in Fig. 2-4) associated with strong positive anomalies in specific humidity (also in relative humidity, Fig. S1.4) and precipitation totals (22 January) as well as negative anomalies in the CRE. These two periods are particularly striking in terms of transport. First, the lower-tropospheric moist anomaly on 22 January, which is related to long-range transport from the upper-level extratropics. Second, the mid-tropospheric moist anomaly on 14 February, which is linked to quasi-horizontal short-range transport from the tropics. In the following Sect. 4 and 5, we have a closer look at these two events, in particular, the air parcel transport histories, to understand the formation of the local anomalies.

## 4   Case study about the link between EDIs and low-level moist anomalies in the trades

This section illustrates how an EDI and its interaction with a trailing cold front induces the lower-tropospheric moist anomaly over Barbados on 22 January 2020. First, we present the Eulerian characteristics of the event (Sect. 4.1), followed by the Lagrangian analysis (Sect. 4.2). In order to put the case study results in a broader context, results from a quasi-climatological assessment of EDI events in Barbados are discussed in Sect. 4.3.

## 4.1 The impact of a trailing cold front on the trade-wind region

On 22 January 2020, a trailing cold front, visible as a cloud band loosely connected to a cyclone over the subtropical North Atlantic (Fig. 5a), reached Barbados. The resulting mesoscale cloud organisation in the vicinity of the island can be described as cloud bands typical for the Fish cloud pattern following the classification of Stevens et al. (2020). The Fish cloud broke up into smaller fragments the day after its passage over Barbados, while travelling further into the tropics (not shown). On 22 January 2020, a saturated layer developed in the early morning near 800 hPa, which grew vertically up to 600 hPa at the expense of the dry air aloft, as can be seen in the profiles from the balloon soundings (Fig. 5b) and the cloud radar's measurements (Fig. 5c). The precipitating humid layer persisted until shortly before local midday (confirmed by personal observation on-site).

The lower-tropospheric moist layer, formed by the passage of the cold front on 22 January 2020, can be clearly identified as strongly anomalous compared to the rest of the campaign in the daily mean $\text{IWV}_{1000\text{-}650\,\text{hPa}}$ (Fig. 6a). The cold front led to prolonged precipitation periods in Barbados such that the precipitation total on 22 January 2020 is also strongly anomalous compared to the rest of the campaign (Fig. 6b). This is consistent with the study by Schulz et al. (2021), who showed that Fish clouds are associated with high rain amounts, but not necessarily high rain rates. Both anomalies, $\text{IWV}_{1000\text{-}650\,\text{hPa}}$ and precipitation totals, are more pronounced in the measurements than in ERA5 (Fig. 6a,b).

The Fish cloud had a net cooling effect over Barbados in the daily mean (Fig. 6c), leading to strongly negative CRE values of about $-38\,\text{W}\,\text{m}^{-2}$ in ERA5 and $-47\,\text{W}\,\text{m}^{-2}$ in CERES on 22 January 2020. This reflects that the cooling shortwave effect during daytime dominated the continuous but comparably weak warming longwave effect (Fig. 3). Note that this effect is spatially confined to the cloud band (Fig. S1.7). If averaged over a $10° \times 10°$ domain (10-20° N, 60-50° W; red box in Fig. 5a), the CRE on 22 January 2020 reduces to values of about $-27\,\text{W}\,\text{m}^{-2}$ in ERA5 and $-8\,\text{W}\,\text{m}^{-2}$ in CERES, which is still anomalously negative compared to the rest of the campaign according to ERA5, but not according to CERES (Fig. 6c). Looking at the spatial distribution of the CRE in the considered domain (see exemplary time step in Fig. S1.7), we note that ERA5 overestimates the presence of liquid/low-level clouds compared to CERES, leading to a stronger cloud radiative cooling in ERA5. Generally, ERA5 shows 10 to 20 $\text{W}\,\text{m}^{-2}$ more negative net cooling over the $10° \times 10°$ domain compared to CERES or the satellite-based study of Bony et al. (2020, their Fig. 5). Deviations of a similar magnitude have been found in other studies comparing the CRE derived from reanalysis data to satellite-based products (Joos, 2019, their Fig. 3 and 4). In the next subsection, we discuss to what extent the cold front's movement into the tropics, and the associated EDI, contribute to the above described anomalous local conditions.

## 4.2 Descent from the extratropical upper troposphere into the trade-wind cloud layer

The trajectories leading into the anomalously moist layer (1000-650 hPa) over the BCO on 22 January 2020 (Fig. 7) can be bundled into five coherent airstreams. To assess these airstreams, we first identify the trajectories with a strong descent and assign them to three EDI airsteams (the leading L in the airstreams' names stands for the arrival in the lower troposphere):

- **L-EDIwcb**: Trajectories with a strong EDI descent ($\geq 400\,\text{hPa}\,(48\,\text{h})^{-1}$), which also experienced a warm conveyor belt (WCB)-like ascent ($\geq 600\,\text{hPa}\,(48\,\text{h})^{-1}$; following Wernli, 1997) from the lowermost to the upper troposphere prior to the EDI descent, identified during the ten days before arrival.

- **L-EDIs**: Trajectories with a strong EDI descent ($\geq 400\,\text{hPa}\,(48\,\text{h})^{-1}$) within ten days before arrival, but no previous WCB-like ascent.

- **L-EDIw**: Trajectories with a weak EDI descent ($\geq 300\,\text{hPa}\,(48\,\text{h})^{-1}$) within ten days prior to arrival.

The remaining trajectories are divided into two airstreams depending on whether they entered the lower troposphere from above or were already in it during the five days before arrival:

- **L-MT**: Trajectories that descend from the mid-tropospheric layer into the lower-tropospheric layer (at a lower rate than the EDI-airstreams) during the five days before arrival.

- **L-LT**: Trajectories that remain in the lower-tropospheric layer during the five days before arrival.

Overall, the transport into Barbados' lower troposphere on 22 January 2020 was influenced by an extratropical upper-level Rossby wave breaking over the central North Atlantic roughly four days earlier (Fig. 7, Fig. S2.4 and animation in Supplement 3), associated with an upper-level PV streamer (red contour in Fig. 7) and an intense surface cyclone underneath (black contour in Fig. 7, using the detection method of Wernli and Schwierz, 2006). A region of dynamically driven large-scale subsidence is found upstream of the PV-streamer (Fig. 7a-c), as expected from quasi-geostrophic theory (Davies and Wernli, 2015). From 17-19 January 2020, the three L-EDI airstreams (L-EDIwcb, L-EDIs, and L-EDIw; brown upward-/downward-facing triangles and rhombi in Fig. 7; brown lines in Fig. 8) were steered into this region at upper levels ($p \sim 500\,\text{hPa}$) and rapidly descended into the cold sector of the extratropical cyclone (Fig. 8a) where they met the L-LT airstream (green left-facing triangles in Fig. 7a).

The descent caused a massive decrease in relative humidity along the trajectories (Fig. 8c), as the L-EDI air parcels were adiabatically warmed while staying dry (Fig. 8f) reflecting limited mixing with the environment. Simultaneously, ocean evaporation beneath the L-EDI air parcels was reinforced (blue contours in Fig. 7b; Fig. 8d). On 20 January 2020 the EDI-like descent was completed and the L-EDI penetrated the cloud layer, which is marked by a sudden increase in relative humidity (Fig. 8c) and liquid water content (Fig. 8e). Once the airstream arrived in the sub-cloud layer, the specific humidity continued to increase (Fig. 8f), while the liquid water content decreased (Fig. 8e), which likely contributed to the L-EDI air parcels' moistening through hydrometeor evaporation. The enhanced surface evaporation starting with the EDI's descent (resulting from the advection of dry unsaturated air over the ocean surface, e.g., Aemisegger and Papritz, 2018) and the subsequent increase in specific humidity approximately 24 h later is typical for EDIs over the North Atlantic (Raveh-Rubin, 2017, their Fig. 8 and 11).

The L-LT air parcels (remaining in the lower-tropospheric layer during the five days before arrival) moved towards lower latitudes along the western edge of the surface cyclone (Fig. 7a) while slightly descending (green line in Fig. 8a). They experienced a gradual increase in relative (green line in Fig. 8c) and specific humidity (Fig. 8f), which set in earlier than for the L-EDI airstreams. On 19-20 January 2020, the L-LT's moistening (Fig. 8f) temporarily flattened due to mixing with the

L-EDI airstreams. At the same time, the PV streamer broke up and a PV cutoff formed northeast of Barbados (Fig. 7c-d). The low-level cyclonic circulation induced by the PV cutoff pushed the L-LT and the L-EDI air parcels (located west of the cutoff) southwards (Fig. 7d). While further descending, the L-LT and L-EDI air parcels caught up with the cold front. The cold front is identifiable in a region with a strong horizontal gradient in equivalent potential temperature (Fig. S2.4) behind a line of high total column water (Fig. 7b-d), which is associated with low-level convergence and precipitation (Fig. S2.5). At roughly $20°$ N, the L-LT and L-EDI air parcels entered the trade-wind region and eventually reached Barbados at low levels (1000-650 hPa).

The remaining L-MT air parcels (slowly descending from above 650 hPa) moved predominantly within tropical latitudes (purple line in Fig. 8b) and descended moderately (Fig. 8a), following a typical Hadley-type transport pathway. This airstream arrived at the top of the lower troposphere in the region of the inversion and most likely mixed only marginally with the cloud layer air, as shown by the low values in liquid water content (Fig. 8e) and specific humidity (Fig. 8f).

In summary, $45\%$ of the 1200 air parcels arriving between 1000 and 650 hPa on 22 January 2020 belong to the L-EDI airstreams (Table 1), contributing to $58\%$ of the $\text{IWV}_{1000\text{-}650\,\text{hPa}}$. The L-LT contains $20\%$ of the trajectories and makes up $26\%$ of the $\text{IWV}_{1000\text{-}650\,\text{hPa}}$ (Table 1). Lastly, L-MT holds $35\%$ of the trajectories, which contributes $16\%$ of the $\text{IWV}_{1000\text{-}650\,\text{hPa}}$ (Table 1). Thus, we conclude that the EDI air parcels were the key players in the formation of the low-tropospheric moist anomaly on 22 January 2020.

### 4.3 The climatological relevance of EDIs for the environmental conditions in the trades

In the following, we assess how frequently EDIs occur in general and how often they are related to local conditions similar to the ones on 22 January 2020. For this, we extract all EDI days from the 11 Januaries and Februaries 2010-2020 (quasi-climatological period). The remaining days are referred to as nonEDI days. An EDI day is identified if at least $5\%$ of the trajectories arriving from 00 to 23 UTC between 1000 and 650 hPa over the BCO descended $400\,\text{hPa}\,(48\,\text{h})^{-1}$ within the four days prior to arrival. The considered time span of four days ensures that only EDIs subsiding over the North Atlantic (corresponding to the extratropical dry intrusion regime in Aemisegger et al. (2021), their Fig. 3a) and not over Europe/Africa (corresponding to the extratropical trade-wind regime in Aemisegger et al. (2021), their Fig. 3b) are selected. Typically, air parcels from the extratropical trade-wind regime require about six days to traverse the North Atlantic. The selection criterion is subjectively chosen but ensures that the well-documented EDI cases on 22 January 2020 and on 31 January 2018[1] are included.

Applying this criterion, we find that EDIs arriving at Barbados are comparatively rare. Only 44 ($\sim$7%) of the total 652 days in the quasi-climatological period are identified as EDI days (Table 2). The local conditions at the BCO on EDI days range from extremely dry with no precipitation to extremely moist with high precipitation totals and varying CRE (Fig. 9). In the considered period, EDIs occur on six ($18\%$) of the 33 driest days according to $\text{IWV}_{1000\text{-}650\,\text{hPa}}$, on eight ($24\%$) of the 33 wettest days according to total precipitation, and on seven ($21\%$) of the 33 days with the lowest CRE. Other (presumably local) processes are responsible for the rest of the three variables' extremes. Thus, even though EDIs arriving in Barbados are

---

[1]Here we only identify the onset of this event because of using a slightly different EDI selection criterion and a smaller set of trajectories. Note that the results' sensitivity to the EDI selection criteria is addressed in Fig. S1.2

relatively rare, they can trigger extremes of either sign in $IWV_{1000\text{-}650\,hPa}$, and they occur in connection with heavy precipitation and anomalously low CRE more frequently than their climatological occurrence frequency indicates.

On EDI days, the two variables, $IWV_{1000\text{-}650\,hPa}$ and precipitation totals, are more strongly positively correlated than on nonEDI days (Table 3). The relation between CRE and precipitation or $IWV_{1000\text{-}650\,hPa}$ is only marginally stronger on EDI days than on nonEDI days. Nevertheless, we observe a shift towards lower daily mean CRE values on EDI days (with a mean of $-28\,\mathrm{W\,m^{-2}}$) compared to nonEDI days (with a mean of $-18\,\mathrm{W\,m^{-2}}$). Especially the EDIs with extremely high $IWV_{1000\text{-}650\,hPa}$ and daily precipitation totals (in the upper right corner of Fig. 9a,b), consistently have an enhanced cloud

radiative cooling. Two processes in connection with EDIs contribute to a strengthening of the cloud radiative cooling. Due to the EDI's subsidence we expect an anomalously dry free troposphere (no mid-/upper-level clouds), which strengthens the longwave radiative cooling (Cau et al., 2007) and explains low CRE on days with a dry lower troposphere. Additionally, the EDI can be associated with a Fish cloud (as on 22 January 2020), which strengthens the shortwave radiative cooling (Bony et al., 2020) and leads to low CRE on days with a moist lower troposphere. Anomalously high CRE on EDI days are most

likely caused by other processes higher up in the atmosphere such as cirrus clouds, which affect CRE but have no impact on the low-level variables. These overall stronger links between precipitation, $IWV_{1000\text{-}650\,hPa}$ and CRE emerge due to the large-scale nature of the perturbations induced by EDIs.

We assume that the difference between the extremely moist and wet EDIs (in the upper right corner of Fig. 9a,b) and the remaining EDIs arises from mesoscale details of how the EDIs modulate the dynamical properties of the associated trailing

cold fronts, which can be best described by the low-level divergence in the region of Barbados (Fig. 9b). As we know from theory, a frontogenetic front is characterised by a strong dipole in low-level divergence associated with the frontal ageostrophic circulation. Behind the cold front, namely in the cold sector in which the EDI spreads out, low-level divergence dominates (see also Supplement 2 for the horizontal and vertical sections across the horizontal divergence field associated with a cold front). Thus, a strongly negative daily mean divergence (i.e., convergence) on an EDI day possibly indicates that Barbados was

heavily influenced by the front's warm side, while a strongly positive value hints towards a prolonged influence of the front's cold side and the following cold sector. To test this hypothesis, we study the transport history of the five EDIs with the most negative and the most positive low-level divergence, referred to as $EDI_{con}$ and $EDI_{div}$, respectively (Fig. 10, listed in Table 2) and the synoptic situation of the two most extreme cases (Fig. 11).

$EDI_{con}$ trajectories descend strongly (Fig. 10a), reach cloud tops 2-3 days before arrival (peak in liquid water content,

Fig. 10c), dive into the boundary layer, and experience an important increase in liquid water content when passing through the front and ascending again on the warm side one day before arrival (see exemplary case in Fig. S2.4-S2.5). Due to their closeness to the surface, $EDI_{con}$ trajectories efficiently trigger ocean evaporation (Fig. 10b), which contributes to a rapid increase in specific humidity (Fig. 10d). Unlike $EDI_{con}$, $EDI_{div}$ trajectories show a slowdown of the descent when reaching $\sim 800\,\mathrm{hPa}$ (Fig. 10a; see exemplary case in Fig. S2.6-S2.7). From then on liquid water content remains high (Fig. 10c), suggesting that the

trajectories stay in clouds and do not reach the sub-cloud layer. Consequently, $EDI_{div}$ trajectories are associated with lower values of surface evaporation (Fig. 10b) and specific humidity (Fig. 10d) than $EDI_{con}$. The synoptic situations on the EDI day with the lowest and highest low-level divergence (and two additional cases discussed) confirm the above described mechanisms.

**Strongest EDI$_{con}$ event:** An event in the EDI$_{con}$ category with strong low-level convergence ahead of a frontogenetic cold front occurred on 27 January 2010 with the following characteristics: divergence = $-15.8 \cdot 10^{-6}\,\mathrm{s}^{-1}$, IWV$_{1000\text{-}650\,\mathrm{hPa}}$ = 33.8 mm, precipitation = 3.2 mm, and CRE = $-29.8\,\mathrm{W\,m}^{-2}$. At 12 UTC 27 January 2010, Barbados was located on the warm side of the cold front in a region of strong low-level convergence (Fig. 11a). The red thick contours in Fig. 11a show ERA-Interim fronts identified with a horizontal gradient in equivalent potential temperature at 850 hPa larger than $3.5\,\mathrm{K}\,(100\,\mathrm{km})^{-1}$ (Schemm et al., 2015). The EDI air parcels in this case moved from the cold sector through the front, most likely due to the ageostrophic circulation associated with the front. This idea is supported by the Q-vectors (arrows in Fig. 11a) which point from low to high equivalent potential temperature near the front, indicating frontogenesis, as expected (see Sect. 1) when a deformation flow (the EDI) acts on a horizontal temperature gradient (the front). Consequently, the ageostrophic circulation at the front is upheld (shown by the neighbouring bands of low-level divergence and convergence in Fig. 11b; note that a different pressure level is shown than in Fig. 11a to enable a direct comparison to Schulz et al., 2021) while it passes Barbados. The dynamics of the front directly impacts the observed local conditions. The low-level convergence leads to an accumulation of moisture increasing the IWV$_{1000\text{-}650\,\mathrm{hPa}}$ value. The ascending motion on the front's warm side promotes cloud formation reducing the shortwave radiative input and potentially initiating precipitation. Strong convergence at 950 hPa was previously linked to the Fish cloud pattern (Schulz et al., 2021) and thus to strong cloud radiative cooling (Bony et al., 2020). Indeed, we find a distinct cloud band directly over/close to Barbados on the five considered EDI$_{con}$ days (exemplary cases in Fig. S2.1).

**Strongest EDI$_{div}$ event:** An event in the EDI$_{div}$ category with low-level divergence behind the cold front occurred on 3 February 2013 with the following characteristics: divergence = $22.4 \cdot 10^{-6}\,\mathrm{s}^{-1}$, IWV$_{1000\text{-}650\,\mathrm{hPa}}$ = 19.2 mm, precipitation = 0 mm, and CRE = $-0.8\,\mathrm{W\,m}^{-2}$. At 06 UTC 3 February 2013 (Fig. 11c), the island was located in the cold sector behind a cold front (Fig. 11c) in an environment of strong low-level divergence (Fig. 11d) leading to the exact opposite characteristics than on 27 January 2010, i.e., low IWV$_{1000\text{-}650\,\mathrm{hPa}}$, cloud-free conditions and no precipitation. However, it would be misleading to relate EDIs arriving together with the cold sector exclusively to cloud-free conditions. Often the cold sector features typical open or closed convection cells (e.g., Krueger and Fritz, 1961; Agee et al., 1973, exemplary cases in Fig. S2.2), which might further explain the large variability of CRE on EDI days.

The main difference between events in the EDI$_{con}$ and EDI$_{div}$ categories is the relationship between the cold front and the EDI. In the first case, the EDI overtakes the cold front by entering the boundary layer on the cold front's cold side and subsequently ascending on the front's warm side. In the second case, the front and the EDI are spatially and temporally separated, with the front propagating faster than the EDI and the EDI entering the boundary layer only when arriving in Barbados. However, the fact that the EDI catches up with the cold front is not sufficient for wet conditions to arise. We found at least one case on 31 January 2018 (discussed in Aemisegger et al., 2021), when the EDI air parcels arrive in Barbados together with the warm air ahead of the cold front, but the distinct dipole structure of low-level divergence at the front had already dissolved (front in a frontolytic state). A plausible cause for these contrasting situations is the upper-level forcing and the corresponding strength of the surface cyclone which is notably stronger for EDI$_{con}$ days than for EDI$_{div}$ (Fig. 11, Fig. S2.3) and therefore supports the stronger deformation and frontogenesis.

In at least two of the analysed cases (Fig. S2.3-S2.5), the trailing cold front in the subtropics (independent on whether they fall into the $EDI_{con}$ or $EDI_{div}$ category) adopt characteristics similar to katafronts (see introduction). The warm and moist subtropical boundary layer air does not ascend along the backward tilted frontal surface (as typical for anafronts), but remains ahead of the surface cold front. As a result, precipitation (if present) falls in a narrow band (the so-called Fish cloud) along the surface cold front.

Overall, we demonstrated that EDIs descending from the midlatitude jet stream region into low latitudes disturb the trade-wind region's environmental conditions such as CRE, precipitation and $IWV_{1000-650\,hPa}$. If the EDI catches up with the cold front, this can lead to a low-tropospheric moist anomaly, the occurrence of the Fish cloud pattern and enhanced radiative cooling. Due to their large-scale flow forcing, EDIs are good candidates for studying the cloud-circulation coupling in detail also in coarse resolution (0.5-1° horizontal grid spacing) climate simulations.

## 5  Case study about the link between TMDs and mid-level moist anomalies in the trades

In this section we analyse an event with TMD (tropical mid-level detrainment) near the melting layer that leads to the mid-tropospheric moist anomaly over Barbados on 14 February 2020 (Fig. 2a). Similar to the first case study, we first discuss the Eulerian (Sect. 5.1) and Lagrangian (Sect. 5.2) characteristics of the event, followed by a quasi-climatological assessment of tropical detrainment at mid levels affecting Barbados (Sect. 5.3).

### 5.1  The impact of a mixed-phase shelf cloud on the trade-wind region

On 14 February 2020, Barbados was covered by a mid-tropospheric shelf cloud detrained from cumulonimbi over South America (shown in Sect. 5.2) forming a large-scale nearly closed altocumulus layer extending from 5° N over South America northwards to about 18° N (Fig. 12a). The large-scale dimension of the mid-level cloud layer is emphasized in the Poldirad cross section (Fig. 12d), where it spans the full width of 120 km. The shelf cloud persisted throughout 14 and 15 February and dissipated on 16 February 2020 (not shown). The saturated layer associated with these mixed-phase clouds (Fig. 13b) was relatively deep and extended between the 0° C isotherm near 600 hPa and 300 hPa at −30° C (Fig. 12b). A distinct dry layer separated the shelf cloud from the shallow trade-wind cumuli beneath (Fig. 12b-d). This dry layer was moistened through the evaporation of hydrometeors falling from the shelf cloud. In some cases the precipitation reached the surface, e.g., 20 km away from the Poldirad (Fig. 12d). In other cases it completely evaporated, as shown in the cloud radar image (Fig. 12c) in the form of downward propagating reflectivity signals that, however, do not reach the shallow trade-wind cumuli. The evaporative cooling in the dry layer most probably helped to maintain a stable layer near the 0° C isotherm (as discussed in e.g., Zuidema et al., 2006).

We assume that this moistening process created favorable conditions for the vertical growth of shallow convection beneath. Indeed, individual convective cells penetrating the dry layer from below are visible from the large radar reflectivity values in the two radar images, e.g., after 20 UTC above Barbados (Fig. 12c) and about an hour earlier between 80 and 85 km away from the Poldirad (Fig. 12d). In these situations enhanced shallow convection injected moisture upward into the shelf cloud,

potentially contributing to the persistence and considerable northward extension of this layer. At least in two cases, at 08 and 14 UTC, such a convective cell pushing through the trade inversion was observed shortly after a moistening event of the dry layer by evaporation of hydrometeors falling out of the altocumulus layer (Fig. 12c). This temporal succession suggests that evaporatively driven downdrafts might have acted as an initiating process for more vigorous shallow convection at low levels (e.g., Zuidema et al., 2012; Li et al., 2014; Vogel et al., 2016). However, we cannot distinguish between the temporal evolution and the advection of clouds (low-level easterlies and mid-level southerlies, Fig. 12b) in the images from the vertically pointing cloud radar (Fig. 12c). Thus, mid- and low-level features that appear in close temporal succession are not necessarily causally linked.

Above Barbados, the shelf cloud caused a positive anomaly in the daily mean of the $\text{IWV}_{650\text{-}300\,\text{hPa}}$ (Fig. 13a) and the total ice water (Fig. 13b) compared to the remaining campaign days. Similar to the first case study, the anomalies are stronger in the measurements than in the reanalysis. Both data sets indicate an enhanced cloud radiative cooling above Barbados and over the $10° \times 10°$ domain (Fig. 13c). The absolute CRE values (at the BCO $-79\,\text{W}\,\text{m}^{-2}$ in ERA5, and $-61\,\text{W}\,\text{m}^{-2}$ in CERES; over the domain $-34\,\text{W}\,\text{m}^{-2}$ in ERA5, and $-19\,\text{W}\,\text{m}^{-2}$ in CERES) are lower in ERA5 than in CERES. This observation fits well with the higher total ice water recorded by CERES and the general overestimation of the cloud radiative cooling by ERA5, already observed in the first case study. In contrast to the first case study, however, the CRE anomaly is also negative over the domain (lowest two boxplots in Fig. 13c). In the following subsection, we examine the transport history of the air parcels that arrive at mid-tropospheric levels and outline the role of tropical dynamics.

## 5.2 Ascent in the ITCZ and detrainment into the trade-wind lower free troposphere

The air parcels that arrived in Barbados in the mid-tropospheric layer (650-300 hPa) on 14 February 2020 originate from different, mainly tropical and subtropical locations (Fig. 14a). To identify the pathways that eventually led to the moist anomaly, we define four airstreams based on the trajectories' pressure evolution (each trajectory is assigned to one airstream only and the leading M in the airstreams' names stands for the arrival in the middle troposphere):

- **M-trades**: Trajectories that enter the mid-tropospheric layer from below.

- **M-local**: Trajectories that remain in the mid-tropospheric layer in the ten days before arrival.

- **M-UT1**: Trajectories that cross the mid-tropospheric layer from below and re-enter it from above.

- **M-UT2**: Trajectories that enter the mid-tropospheric layer from above.

The M-trades air parcels started their journey near the African West coast (green left-facing triangles in Fig. 14a) and crossed the North Atlantic along the southern edge of a persistent anticyclone, centred at roughly $30°\,\text{N}$. They travelled at low altitudes until reaching South America (Fig. 14b), where they rapidly ascended (green line in Fig. 15a) in deep convective systems associated with the ITCZ (shown by the low cloud temperatures in Fig. 14d). The ascent is associated with a drop in specific humidity (Fig. 15e), resulting from cloud formation. Near the melting layer (red dashed line in Fig. 16), the air parcels were detrained and spread northwards embedded in a developing shelf cloud (Fig. 16a). At the cloud's northern tip, some air parcels

left the cloud and penetrated the dry layer beneath (Fig. 16b), presumably due to the subsidence induced by radiative cooling (Stevens et al., 2017, their Fig. 16). Consequently, the airstream's liquid water content temporarily decreased (Fig. 15c) until clouds formed again due to the outflow from cumuli congesti over South America on 12 February 2020 (Fig. 15c, 16c). The specific humidity remained largely unaffected by this process (Fig. 15e). However, these nearly constant values of specific humidity became increasingly anomalous as the M-trades airstream moved northwards into climatologically drier regions (Fig. 15f).

Shortly before arrival in Barbados, ice (Fig. 16d) and snow formed likely due to ascent along the slightly tilted isentropes. The air parcels ascended well above the $0°$ C isotherm, which might have initiated heterogeneous freezing (Fig. 16d; by meridionally shifting the vertical cross section (not shown), we checked that the cloud is a large-scale shelf cloud and not just the upper part of a tilted deep convective tower which intersects the cross section). Simultaneously, the cloud was precipitating into the dry layer beneath (blue dashed contours in Fig. 16d). The distinct pattern of strong downward winds (orange contours in Fig. 16d, already noted in Fig. 2c) below a precipitating cloud points towards evaporatively driven downdrafts, supporting the hypothesis formulated in Sect. 5.1.

The M-local airstream was already located over South America in the mid-tropospheric layer ten days prior to its arrival (purple right-facing triangles in Fig. 14a). Around 10 February 2020, M-local merged with M-trades (green and purple lines in Fig. 15a,b). After that, the two airstreams show very similar transport histories and therefore we refer back to the detailed description in the preceding paragraphs. Together, M-local and M-trades contain $76\%$ of the air parcels that arrive at mid-tropospheric levels on 14 February 2020 and brought $93\%$ of the $IWV_{650\text{-}300\,hPa}$ (Table 4).

Therefore, the remaining airstreams (M-UT1, M-UT2) only contributed little in terms of air parcels and moisture content (Table 4). The few M-UT1 air parcels rose over the eastern Pacific (light brown upward-facing triangles in Fig. 14) from 8-10 February 2020 (light brown line in Fig. 15a). The rapid ascent brought them to subtropical latitudes (Fig. 15b) and resulted in a substantial decrease in specific humidity (Fig. 15e). When the M-UL1 airstream finally arrived in Barbados, it had dried out almost completely (Table 4, Fig. 15e).

The M-UL2 airstream exited the subtropical jet and approached from upper levels slightly north of Barbados or featured a Hadley cell-like descent at tropical latitudes (dark brown downward-facing triangles in Fig. 14). The trajectories in the subtropical jet completed almost a full circle around the globe within ten days, starting their journey over continental North Africa (Fig. 14a). As M-UL2 stayed at upper levels (dark brown line in Fig. 15a), its water content (Fig. 15c-e) remained close to zero throughout the ten days. M-UL1 and M-UL2 make up $24\%$ of the air parcels and hold $7\%$ of the $IWV_{650\text{-}300\,hPa}$ (Table 4). Thus, even though the two airstreams happened to arrive in the mid-tropospheric layer on 14 February 2020, they were not relevant for the observed moist anomaly. In other words, the mid-level moist anomaly above Barbados was essentially produced by outflow from the tropical shelf cloud, which was fed by airstreams from the North Atlantic trades and tropical South America.

## 5.3 The climatological relevance of TMDs for the environmental conditions in the trades

The characteristic features of the most relevant airstreams (M-trades, M-local) involved in the formation of the mid-tropospheric moist anomaly above Barbados on 14 February 2020 are the advection from low latitudes and the detrainment at mid-tropospheric altitudes. To asses the importance of this transport pathway, we select all days from the quasi-climatological period on which at least 25 % of the trajectories arriving 00-23 UTC between 650 and 300 hPa over the BCO travelled at tropical latitudes (minimal latitude $< 10°$ N) and never overshot the arrival layer (minimal pressure $> 300$ hPa) within the ten days before arrival. The latitudinal criteria is chosen based on the climatological position of the ITCZ (in the vicinity of which mid-level detrainment might take place), which is south of $10°$ N in January and February according to Waliser and Gautier (1993, their Fig. 4g). We consider the full ten day time span of the trajectories, because we assume that horizontal transport at mid-tropospheric levels is slow compared to other levels (due to the change from low-level easterlies to upper-level westerlies). The TMD selection criteria are subjective, but serve the purpose to identify days with a transport history similar to 14 February 2020. Note that the demanded percentage of trajectories meeting the criteria for a tropical detrainment day is higher than for an EDI day (Sect. 4.3), because the M-trades and M-local airstreams in Sect. 5.2 contained more trajectories than the L-EDI airstreams in Sect. 4.2. The TMD (and EDI) frequency during the quasi-climatological period for other percentage thresholds can be studied in Fig. S1.2.

Applying this definition, we identify 65 TMD days ($\sim$10 % of the quasi-climatological period), which however belong to fewer events, which typically last for several consecutive days (Table 5). Most (53) TMDs lead to an anomalously moist middle troposphere above Barbados, occasionally associated with the formation of ice crystals (Fig. 17b; the two days with the highest total ice water are 14 and 15 February 2020). Twenty (61 %) of the TMDs belong to the 33 days with the highest values of $IWV_{650\text{-}300\,hPa}$ (Fig. 17a) and 17 (52 %) to the 33 days with the highest total ice water (Fig. 17b). It appears that the mid-tropospheric humidity conditions do not strongly influence the CRE over Barbados (Fig. 17a, Table 6). Thus, the case observed on 14 February 2020 during EUREC[4]A represents a very rare outlier with a CRE of $-79$ W m$^{-2}$.

The TMD trajectories leading to the few (12) dry anomalies ($TMD_{dry}$) in the middle troposphere start at and ascend to lower pressure levels/higher altitudes (Fig. 18a) than the ones leading to moist anomalies ($TMD_{moist}$). Another characteristic of $TMD_{dry}$ trajectories is their excursion to comparably high latitudes (Fig. 18b), which matches one possible formation pathway of mid-tropospheric dry layers over the North Atlantic, described earlier by Casey et al. (2009). One might argue that these cases do not qualify as TMDs, because their ascent and detrainment (if at all) happened earlier than the ten days before arrival. Nevertheless, we can learn from them that for the formation of a mid-tropospheric moist anomaly (through a TMD), it is crucial that the air parcels start at low altitudes (Fig. 18a), where specific humidity is higher (Fig. 18c), and are lifted only as far as the cooling and subsequent condensation does not dry them out completely. The lower the altitude at which the air parcels are detrained, the moister do they arrive in Barbados (Fig. 17; note the high correlation between $IWV_{650\text{-}300\,hPa}$ and minimal pressure).

The quasi-climatological analysis linked tropical mid-level detrainment to mid-tropospheric moist anomalies above Barbados. We established a strong relation between the detrainment height and the magnitude of the moist anomaly. However, the

amount of water vapour between 650 and 300 hPa appeared at large to be unrelated to CRE. The lack of a clear link between mid-tropospheric moisture/clouds fits well with existing literature, showing that the CRE of mid-level clouds depends on the cloud phase (Sassen and Wang, 2012) and thickness of the cloud, as well as the underlying surface and additional cloud layers at different altitudes (Bourgeois et al., 2016).

## 6   Summary and conclusion

In this paper we investigated the formation pathways of moist anomalies in the North Atlantic winter trades during the EUREC[4]A field campaign. We combined observational data with a detailed trajectory analysis based on ERA5 reanalysis data. Starting with a temporal overview of the atmospheric profiles above Barbados (Sect. 3), we found that the local conditions, as well as the large-scale circulation varied substantially during EUREC[4]A. For the most part, anomalies in the Eulerian framework were co-located with anomalies in the Lagrangian framework, which indicates that local conditions were strongly influenced by the transport history of the air. To investigate the link between the air parcels' transport to Barbados and the formation of local moist anomalies, we conducted two detailed case studies of days from EUREC[4]A with moist anomalies in the lower (1000-650 hPa) and middle (650-300 hPa) troposphere, respectively. Both days were associated with exceptionally negative values of CRE.

The first case study (Sect. 4), on 22 January 2020, with a lower-tropospheric moist anomaly over Barbados is associated with a precipitating cloud band of a trailing cold front (a so-called Fish cloud following Stevens et al., 2020) that extended from about $30°$ N to Barbados and induced a locally strong cloud radiative cooling. The largest contribution to the moist anomaly came from air parcels following the pathway of an EDI (extratropical dry intrusion). The EDI and its interaction with the cold front were essential for the formation of the lower-tropospheric moist anomaly. Our analysis has shown that these air parcels experienced three moistening processes that plausibly only took place because of the presence of the EDI and the cold front: (1) the enhancement of ocean evaporation due to the strong vertical humidity gradient created by the EDI's subsidence and penetration into the (sub-)cloud layer, (2) the convergence of moisture and the formation of convective precipitation at the front, and (3) the evaporation of hydrometeors in the frontal region.

Despite EDIs arriving as far south as Barbados are relatively rare ($\sim$7 % occurrence frequency over the 11 Januaries and Februaries in 2010-2020), they are usually associated with extremes of $\text{IWV}_{\text{1000-650hPa}}$ of either sign. Furthermore, they occur together with sustained heavy precipitation and anomalously low CRE more frequently than their climatological occurrence frequency indicates. Either EDIs arrive in Barbados embedded in the cold sector associated with a dry lower troposphere and negligible precipitation, or they arrive together with the front leading to a humid lower troposphere, increased precipitation and an enhanced cloud radiative cooling. Eight EDIs contribute to the 33 days with the highest daily precipitation totals of the quasi-climatological period, but only three to the 33 moistest days. Thus other transport patterns and processes are responsible for most of the highest values of lower-tropospheric specific humidity in the quasi-climatological period. Note that the occurrence frequency of EDIs is highly dependent on the EDI selection criteria, while the link between EDIs and local conditions in Barbados remains the same.

A follow-up study will look into the EDI's interaction with the cumulus cloud deck of the marine boundary layer behind the cold front and the environment in which EDI trajectories penetrate through the cloud-capping inversion. Furthermore, the interaction between the cold front and the EDI should be studied on a process level to understand whether the EDI is critical for the propagation of the front and the cold sector into the tropics. For this, it would be insightful to adopt a front-centered perspective by comparing fronts appearing as individual features to fronts which occur together with an EDI. Previous knowledge on this topic exists. Catto and Raveh-Rubin (2019) found that trailing cold fronts at low latitudes in most cases coincide with EDIs. Raveh-Rubin and Catto (2019) showed that EDIs significantly influence the characteristics of the fronts and their surroundings. Lastly, Esler et al. (2003) suggested a link between the surface cyclone's strength and the EDI's penetration depth into the boundary layer and degree of moistening. Additional insights could be gained from the analysis of stable water isotopes which provide information about moisture cycling in frontal regions (Aemisegger et al., 2015; Graf et al., 2019). Expanding the analysis beyond the geographical location of Barbados and the two months of the EUREC$^4$A experiment would increase the number of EDIs and thereby further advance our understanding of their impact on the trade-wind region.

The second case study (Sect. 5), on 14 February 2020, with a mid-tropospheric moist anomaly over Barbados is related to a horizontally extended mixed-phase shelf cloud that persisted over two days. This event was associated with the most negative CRE value during EUREC$^4$A. As observed by radar, precipitation from the shelf cloud mostly evaporated/sublimated in the dry layer below, possibly initiating the formation of downdrafts due to evaporative cooling, thereby potentially triggering enhanced shallow convection from the surface. We hypothesise that this mechanism can feed moisture back into the mid-tropospheric cloud layer, thereby contributing to its substantial northward extension and persistence. The complex interaction between the two cloud layers and the separating dry layer remains to be disentangled. Potentially stable water isotopes could help to understand the involved moist processes (e.g., Risi et al., 2019, 2020). The case study's moist anomaly was caused by air parcels that previously were detrained from cumulinimbi in the tropics. The pre-conditioning (the stable layer at the melting layer) for this TMD (tropical mid-level detrainment) came about due to a sharp vertical gradient in latent heating in tropical deep convective systems. With back-trajectories we showed that the case study's moist anomaly is directly linked to the convective activity over tropical South America.

In a quasi-climatological analysis, we identified an occurrence frequency of TMDs of 10 % over the 11 Januaries and Februaries in 2010-2020. On 82 % of the TMD days, they were associated with anomalously humid conditions in the middle troposphere. Twenty TMDs contribute to the 33 moistest days and 17 to the 33 days with the highest total ice water. Even though 14 February 2020 showed that mid-tropospheric clouds trigger a strong short- and longwave radiative response, the daily mean CRE could not generally be linked to the mid-tropospheric moist anomalies. For future studies, which aim at understanding the role of mid-level clouds for the CRE, we recommend to investigate further aspects than the anomaly in humidity and ice content, such as the microphysical properties or the formation timing of the cloud with respect to the daily cycle. Also for TMDs, the frequency of events is highly dependent on the TMD selection criteria. The link between TMDs and the mid-tropospheric moist anomaly, however, becomes stronger the more TMD trajectories per day are required for a TMD identification. Similar to the quasi-climatological analysis of EDIs (Sect. 4.3), the understanding of TMDs and their impact

on the trade-wind region would further profit from expanding the regional and seasonal foci beyond the BCO and the months January and February.

The detailed analysis of two case studies from EUREC[4]A and the accompanying quasi-climatological investigations have shown that extratropical and tropical dynamics periodically disturb the trade-wind region by inducing large-scale transport patterns that deviate from the usual low-level easterly and upper-level westerly flow regime. On the one hand, EDIs can cause the transport of air parcels from the extratropical upper troposphere to the lower troposphere near Barbados, where they either lead to anomalously dry or anomalously humid conditions associated with a locally enhanced cloud radiative cooling. On the other hand, deep convection in the ITCZ over South America can cause a stable layer near the $0°$ C-isotherm that promotes TMD and northwards transport, eventually leading to moist anomalies in the middle troposphere of Barbados, which, however, are not systematically linked to the CRE. Thus, the large-scale circulation and dynamical processes taking place remotely from the trade-wind region must be taken into account for the understanding of local humidity conditions. Moreover, our analysis highlights the great potential of combining the Eulerian and the Lagrangian perspectives based on reanalysis data with local observations.

*Data availability.* The ERA5 reanalyses are provided by the ECMWF and can be downloaded from the official website (https://www.ecmwf.int/en/forecasts/datasets/reanalysis-datasets/era5). The MODIS Terra satellite images of the Earth Observing System Data and Information System (EOSDIS) are available in the Worldview Snapshots application (https://wvs.earthdata.nasa.gov). The CERES data set was downloaded from the following webpage: https://asdc.larc.nasa.gov/project/CERES/CER_SYN1deg-1Hour_Terra-Aqua-MODIS_Edition4A. The GOES-16 satellite images can be retrieved from the GIBBS imagery service (https://www.ncdc.noaa.gov/gibbs/). The measurements from the Vaisala WXT-520 meteorological ground station (https://doi.org/10.25326/54), the atmospheric soundings (https://doi.org/10.25326/137), and the Poldirad (https://doi.org/10.25326/218) can be obtained from the AERIS data repository. The cloud radar data is available on the BCO's official website (https://barbados.mpimet.mpg.de/).

*Author contributions.* LV performed most of the analysis, with help from FA, and wrote the paper. All authors added to the interpretation and presentation of the results.

*Competing interests.* The authors declare that they have no conflict of interest.

*Acknowledgements.* We thank Marina Dütsch (University of Vienna), Elisa Spreitzer (ETH Zürich), and Sara Müller (ETH Zürich) for scientific discussions on dry intrusions; Stefan Bühler (University of Hamburg) for an inspiring discussion about elevated moist layer on a EUREC[4]A campaign preparation workshop; Michael Sprenger (ETH Zürich) for the technical support; Sandrine Bony (Sorbonne University) and Bjorn Stevens (Max Planck Institute for Meteorology) (and other people involved) for the EUREC[4]A initiative; Claudia Stephan

(Max Planck Institute for Meteorology) for coordinating the atmospheric sounding component of EUREC[4]A; the Max Planck Institute for Meteorology, the Caribbean Institute for Meteorology and Hydrology, the Museum of Barbados, and, in particular, Friedhelm Jansen (Max Planck Institute for Meteorology) and Mario Mech (University of Cologne) for the operation of the BCO. The authors acknowledge the ECWMF for the availability of the ERA5 reanalyses, the Earth Observing System Data and Information System (EOSDIS) for the satellite imagery from the Worldview Snapshots application, NASA for providing the CERES data set, the GIBBS imagery service and all those those who were involved in gathering the observational data used in this publication. We are grateful to the two anonymous reviewers for their constructive comments.

*Financial support.* This project was financially supported by the Swiss National Science Foundation Grant No. 188731. EUREC[4]A was funded with support of the European Research Council (ERC), the Max Planck Society (MPG), the German Research Foundation (DFG), the German Meteorological Weather Service (DWD) and the German Aerospace Center (DLR). MB received funding from the European Research Council, H2020 research and innovation program (INTEXseas, grant no. 787652).

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

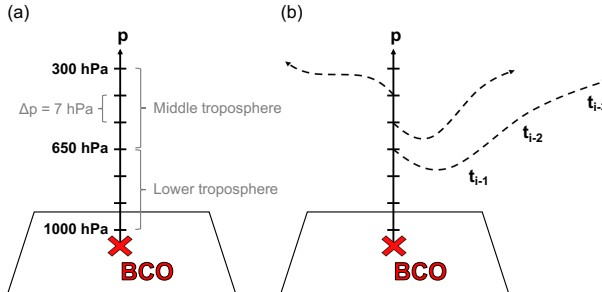

**Figure 1.** Illustrative schematics for **(a)** the definition of the vertical profiles above the BCO and **(b)** the calculation of the three-dimensional backward trajectories.

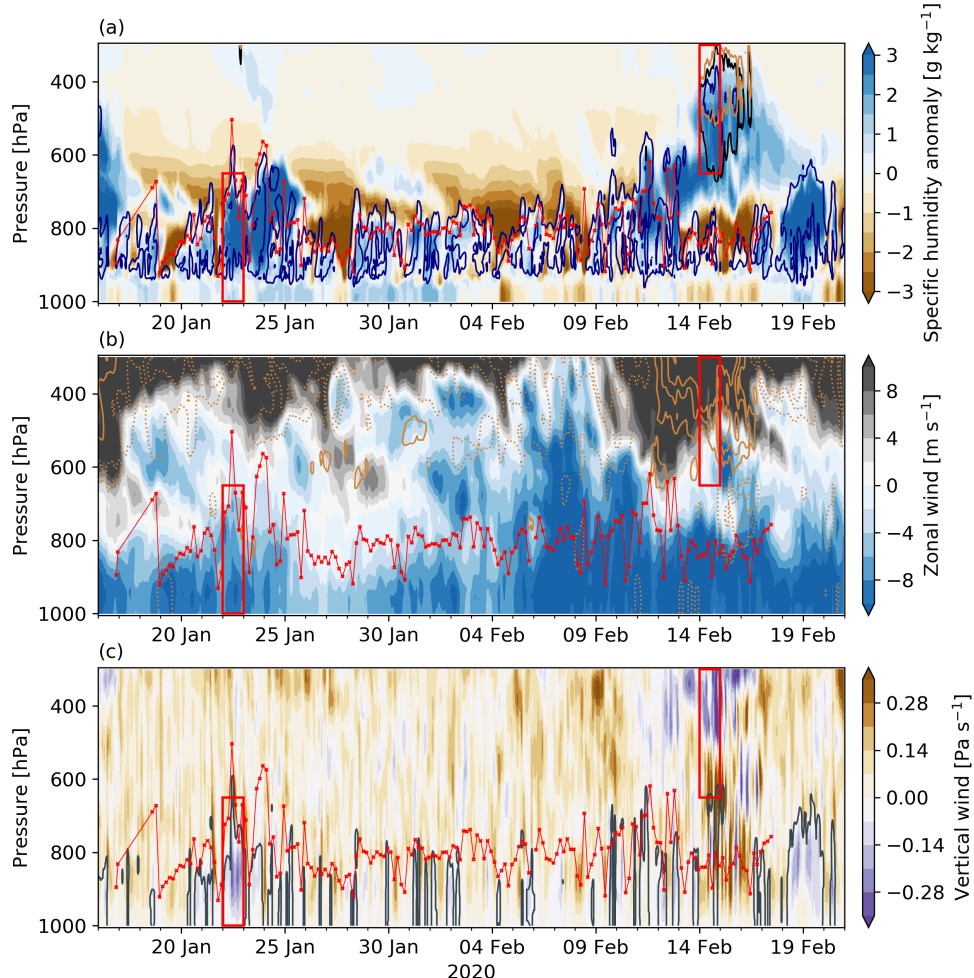

**Figure 2.** Time series of vertical profiles from ERA5 of the environmental conditions at the BCO during EUREC[4]A and height of the lowest temperature inversion (of at least $0.4\,°C$ over the depth of the inversion) derived from the BCO soundings (red lines with dots). Shown are **(a)** the anomaly in specific humidity relative to the level-specific campaign mean, liquid (blue contour), snow (black contour) and ice (brown contour) water content ($10\,mg\,kg^{-1}$), **(b)** the zonal (color shading) and meridional wind ($5, 10, 15\,m\,s^{-1}$, dotted lines for negative values), and **(c)** vertical wind (color shading) and rain water content (black contour for $1\,mg\,kg^{-1}$). A Gaussian filter was applied to all variables for better readability. The red boxes highlight the layers and periods of the two case studies. The time axis is give in UTC (Barbados local time = UTC-4h).

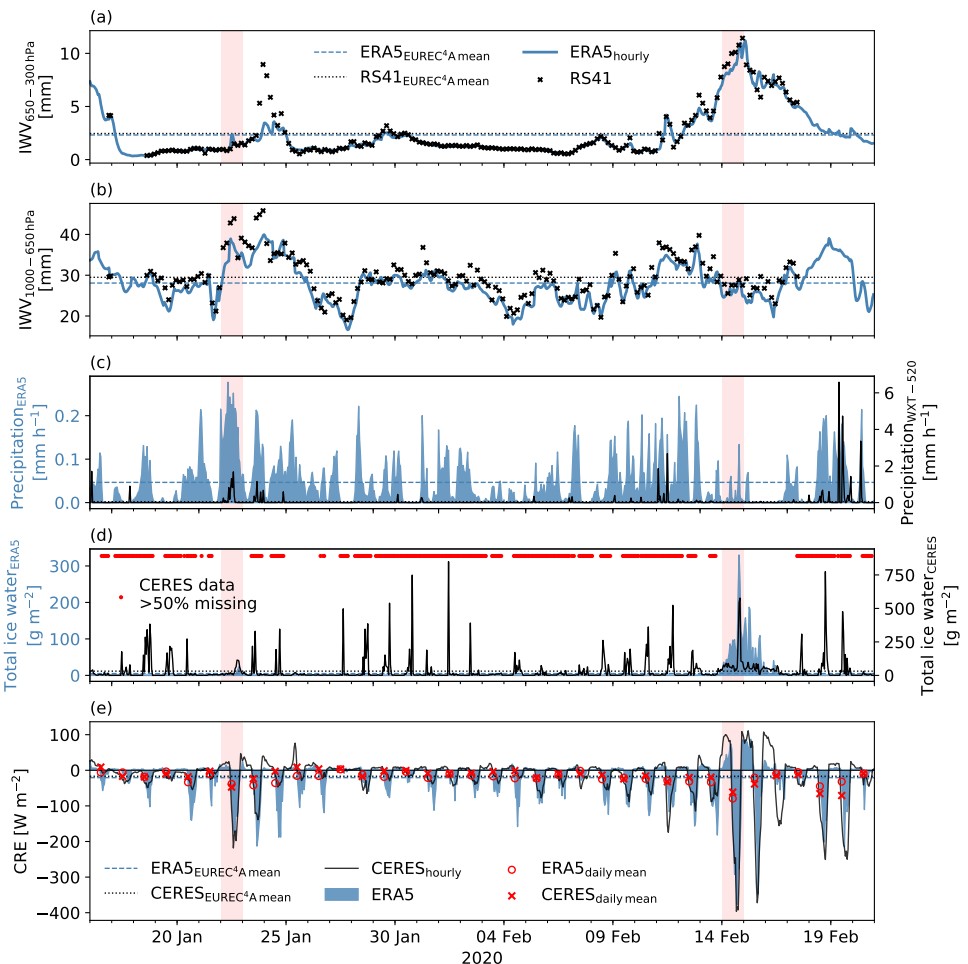

**Figure 3.** Time series of the environmental conditions at the BCO during EUREC[4]A. Shown are **(a)** $IWV_{650\text{-}300\,hPa}$ and **(b)** $IWV_{1000\text{-}650\,hPa}$ from ERA5 and the BCO soundings, **(c)** hourly precipitation totals from ERA5 and the WXT-520 weather station (note the differently scaled y axes), **(d)** total ice water from ERA5 and CERES (note that missing data in CERES was filled with linear spatial interpolation; if more than 50% of the data points in a $10° \times 10°$ domain centered around the BCO were missing, the time step is marked with a red dot at the panel top), and **(e)** CRE from ERA5 and CERES. The red shadings highlight the periods of the two case studies, i.e., 22 January and 14 February 2020. The time axis is give in UTC (Barbados local time = UTC-4h).

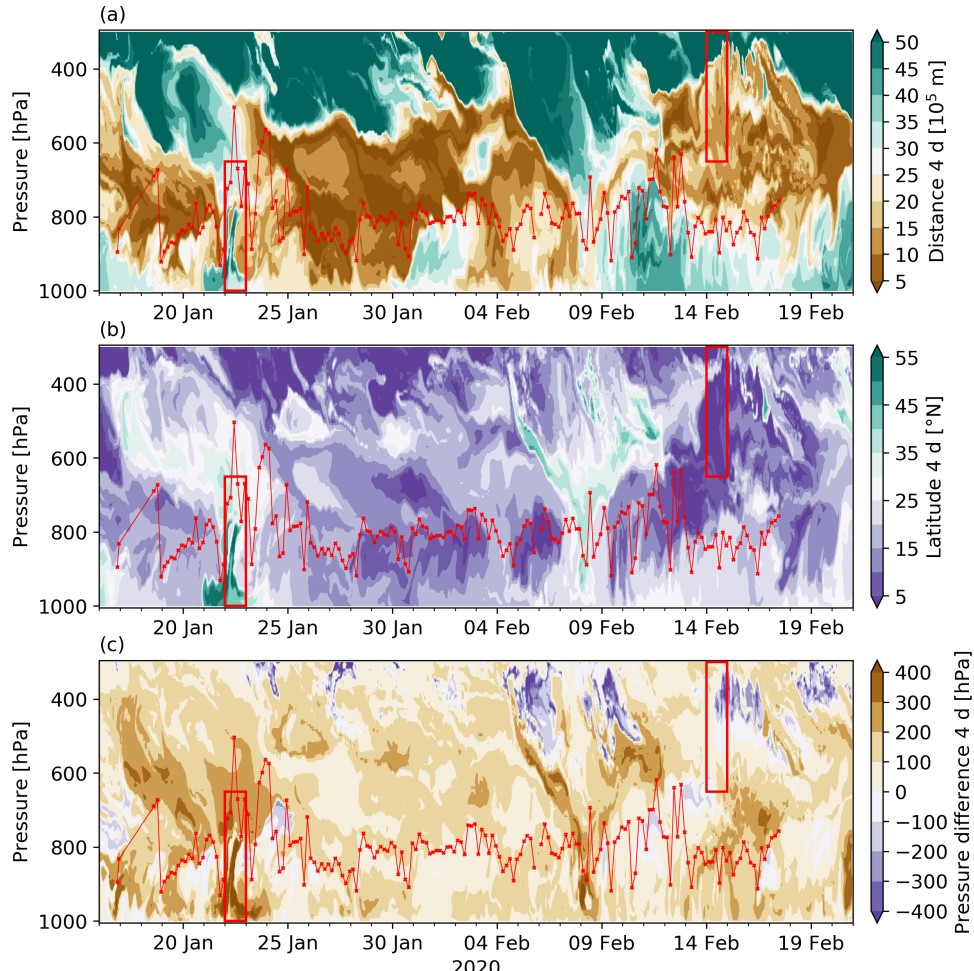

**Figure 4.** Time series of vertical profiles from ERA5 of the transport conditions at the BCO during EUREC[4]A and height of the lowest temperature inversion (of at least $0.4\,^{\circ}\mathrm{C}$ over the depth of the inversion) derived from the BCO soundings (red lines with dots). Shown are the air parcel's **(a)** great-circle distance from the arrival location travelled in the four days prior to arrival, **(b)** latitudinal position four days prior to arrival, and **(c)** pressure change along the trajectory during the four days prior to arrival (positive/negative values indicating descent/ascent towards the arrival location). A Gaussian filter was applied to all variables for better readability. Again, the red boxes highlight the layers and periods of the two case studies. The time axis is give in UTC (Barbados local time = UTC-4h).

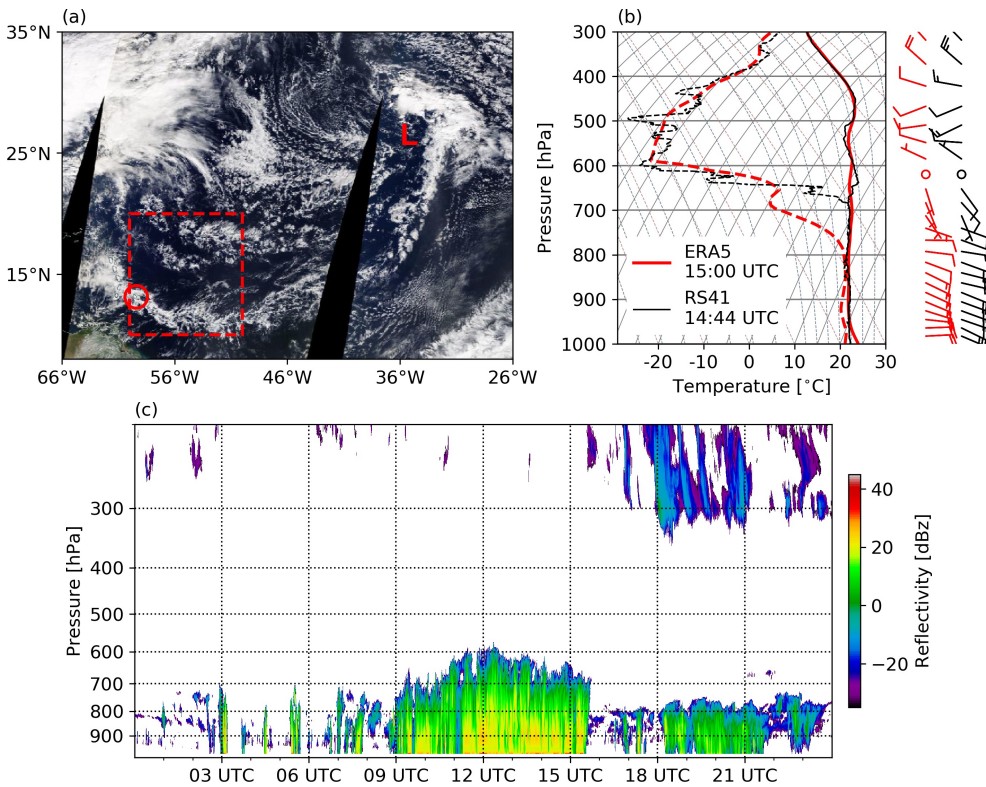

**Figure 5. (a)** MODIS Terra satellite image at about 14:30 UTC 22 January 2020 in the domain 8-35° N, 26-66° W, the location of Barbados (red circle), and the domain used for the domain-mean CRE calculation (10-20° N, 50-60° W; red box). **(b)** Vertical profiles of temperature (solid lines), dew point temperature (dashed lines), and wind (arrows) from ERA5 (red thick lines) and the atmospheric sounding (black thin lines) at about 15 UTC 22 January 2020. **(c)** Vertical profiles measured by the cloud radar (equivalent radar reflectivity of all hydrometeors) at the BCO on 22 January 2020.

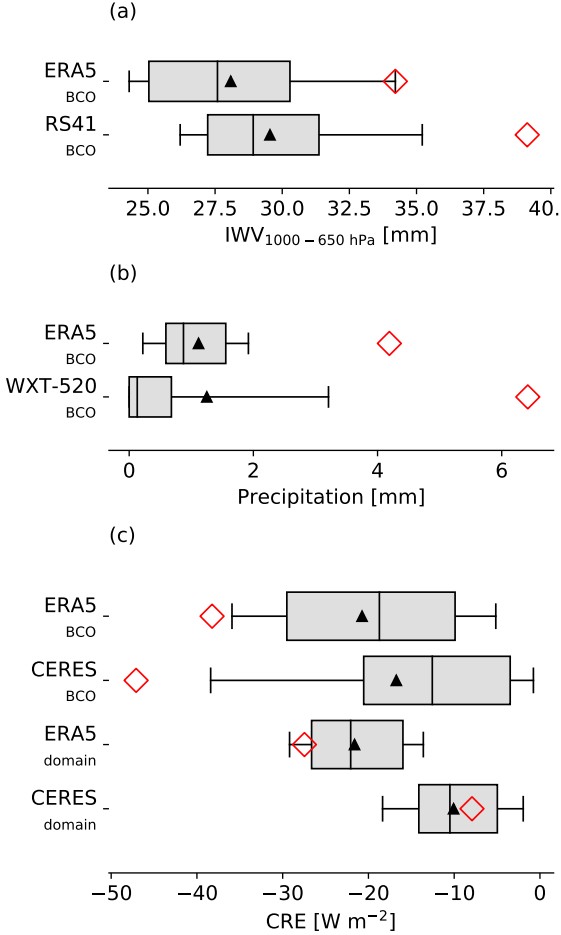

**Figure 6.** Boxplots of **(a)** daily mean $IWV_{1000\text{-}650\,hPa}$ from ERA5 and the atmospheric soundings (RS41) at the BCO, **(b)** daily precipitation totals from ERA5 and the weather station (WXT-520) at the BCO, and **(c)** daily mean CRE from ERA5 and CERES at the BCO and as area-weighted mean over the domain 10-20° N, 50-60° W (red box in Fig. 5a; ERA5$_{domain}$ and CERES$_{domain}$). The boxplots (25, 75 percentiles as box; 10, 90 percentiles as whiskers; median as line in the box, mean as black triangle, outliers not shown) show values of 36 days (16 January to 20 February 2020). The values on 22 January 2020 are shown as red rhombi.

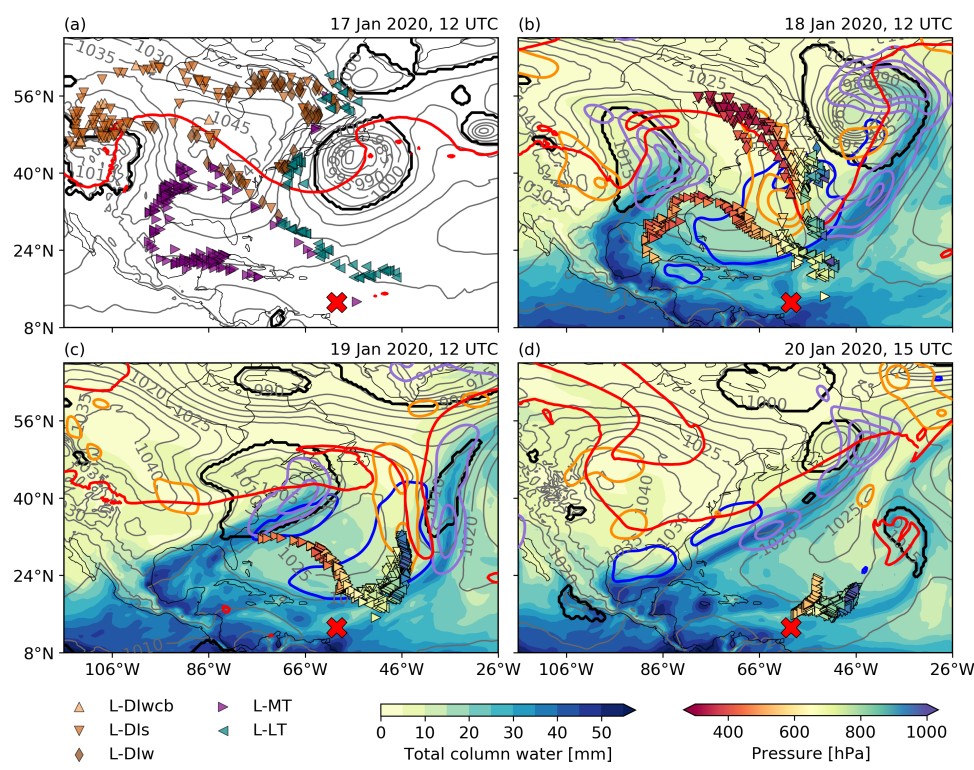

**Figure 7.** Synoptic situation over the North Atlantic and the position of backward trajectories from the BCO started in the layer 1000-650 hPa on 22 January 2020 (00-21 UTC, every 3 h) at **(a)** 12 UTC 17 January, **(b)** 12 UTC 18 January, **(c)** 12 UTC 19 January, and **(d)** 15 UTC 20 January 2020. Trajectory positions are shown as triangles coloured according to the **(a)** airstream or **(b-d)** pressure. Color shading shows total column water, and contours show sea level pressure (grey; 5 hPa intervals), cyclone masks (black), surface evaporation (blue; intervals of 0.5 mm h$^{-1}$), upward and downward winds at 500 hPa (purple and orange, respectively; $\pm 0.2$, $\pm 0.4$, $\pm 0.6$ Pa s$^{-1}$), and 2 pvu at 320 K (red). The red cross marks the location of Barbados. A Gaussian filter was applied to surface evaporation and vertical winds for better readability. Data: ERA5.

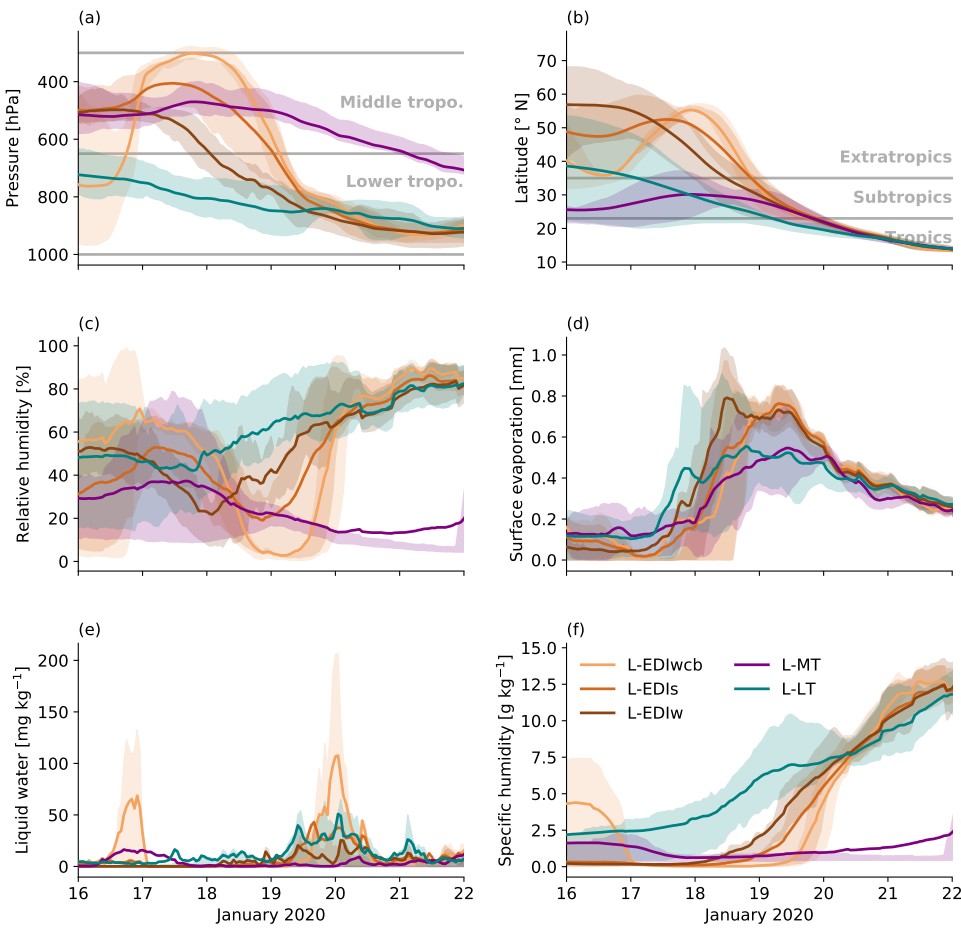

**Figure 8.** Time series along the five airstreams arriving at the BCO in the layer 1000-650 hPa on 22 January 2020. The variables shown are **(a)** pressure, **(b)** latitude, **(c)** relative humidity, **(d)** surface evaporation, **(e)** liquid water content, and **(f)** specific humidity. Thick lines show mean values and the shading the 25-75-percentile range. The number of trajectories assigned to each airstream is given in Table 1. Data: ERA5.

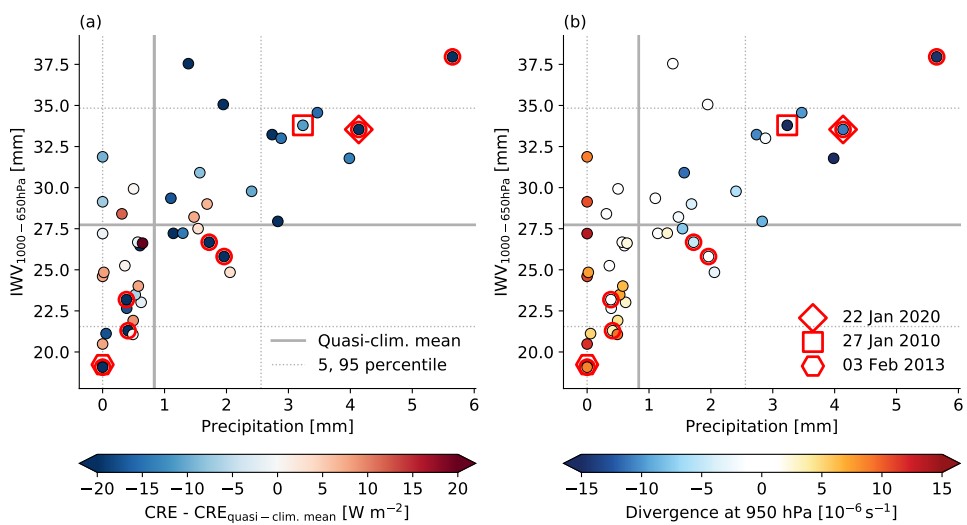

**Figure 9.** The relation between daily mean $IWV_{1000\text{-}650\,hPa}$, daily precipitation totals and **(a)** the anomaly in daily mean CRE relative to the quasi-climatological mean or **(b)** daily mean low-level divergence at the BCO is shown for the 44 EDI days. The quasi-climatological means are shown as gray continuous lines, and the 5 and 95 percentile as gray dashed lines. The red geometrical shapes indicate three specific dates, while the points with a red thick edge belong to the 33 days with the strongest cloud radiative cooling. Note that the daily mean values are based on three hourly data, thus the values are not directly comparable to Fig. 6. Data: ERA5.

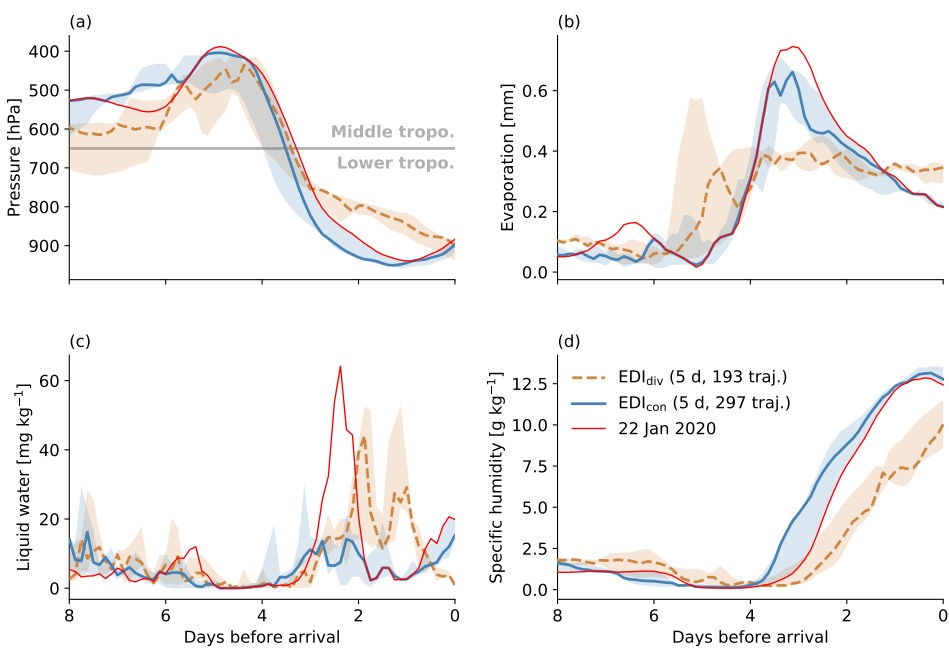

**Figure 10.** Time series along the EDI trajectories arriving at the BCO in the layer 1000-650 hPa during the quasi-climatological period. The variables shown are **(a)** pressure, **(b)** surface evaporation, **(c)** liquid water content, and **(d)** specific humidity. The thick lines show the mean and the shadings the 25-75-percentile range of the trajectories from the five EDI days with the strongest convergence/divergence (blue continuous/brown dashed lines). The mean of the EDI trajectories arriving on 22 January 2020 is shown as red thin line. Data: ERA5.

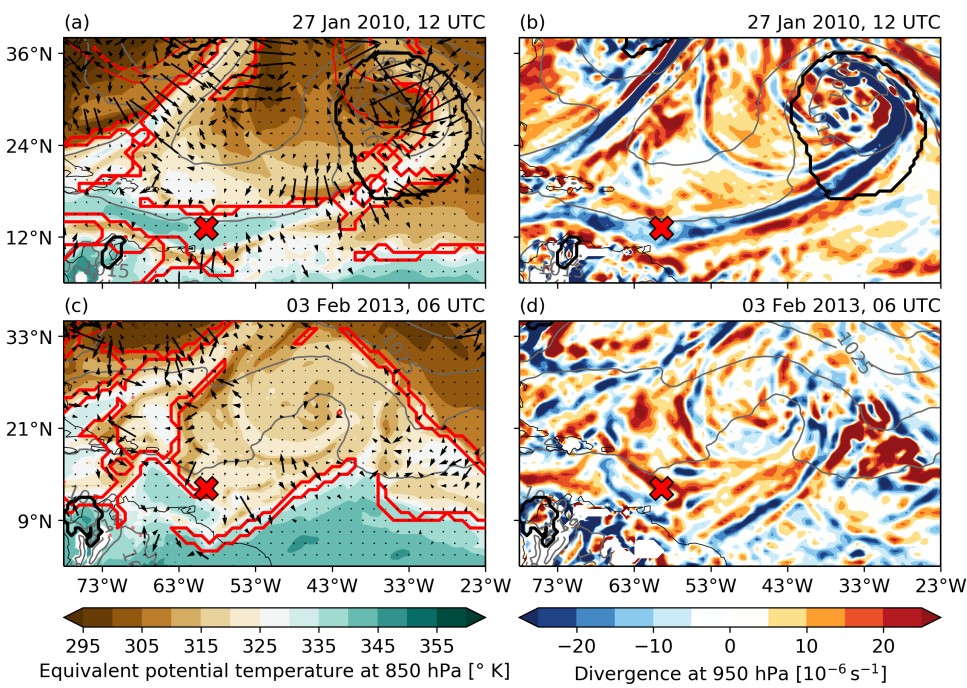

**Figure 11.** Synoptic situation over the North Atlantic on the EDI day with the strongest **(a,b)** low-level convergence (27 January 2010) and **(c,d)** low-level divergence (3 February 2013) in the quasi-climatological period. Shown are **(a,c)** equivalent potential temperature at 850 hPa (shading), ERA-Interim fronts (red thick contours), Q-vectors (calculated with the full horizontal wind at 850 hPa previously smoothed with a Gaussian filter), 2 pvu at 320 K (red thin contours); **(b,d)** horizontal divergence at 950 hPa (shading); sea level pressure (gray; 5 hPa intervals), cyclone masks (black), and the location of the BCO (red cross). Data: ERA5.

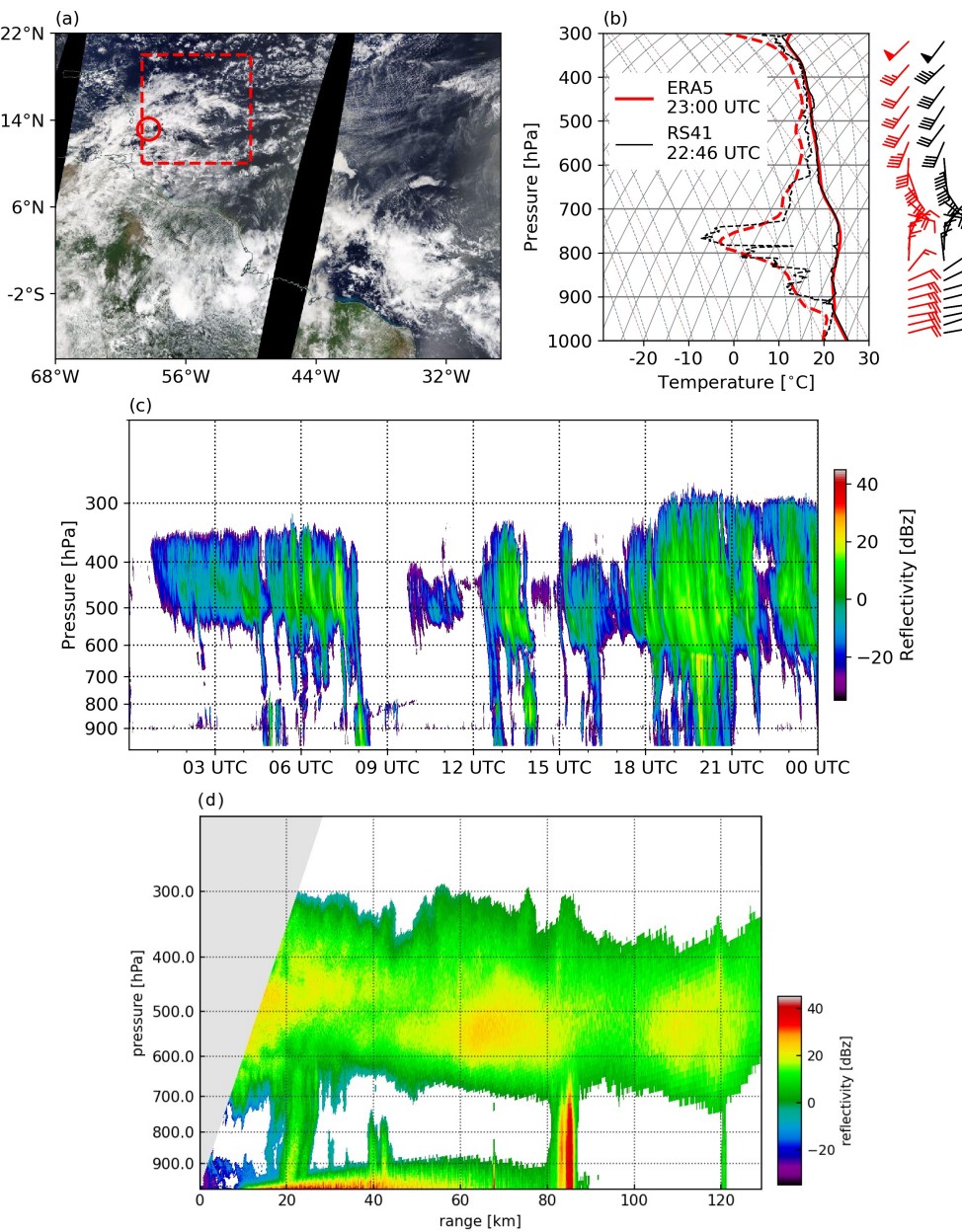

**Figure 12. (a)** MODIS Terra satellite image at about 14:30 UTC 14 February 2020 in the domain 8° S-22° N, 27-68° W, the location of Barbados (red circle), and the domain used for the domain-mean CRE calculation (10-20° N, 50-60° W; red box). **(b)** Vertical profiles of temperature (solid lines), dew point temperature (dashed lines), and wind (arrows) from ERA5 (red thick lines) and the atmospheric sounding (black thin lines) at about 23 UTC 14 February 2020. **(c)** Vertical profiles measured by the cloud radar (equivalent radar reflectivity of all hydrometeors) at the BCO on 14 February 2020. **(d)** Vertical cross section measured by the Poldirad at 19:15 UTC on 14 February 2020 when the instrument was facing towards the BCO. Note, the shallow reflectivity layer between 10 and 90 km is sea clutter.

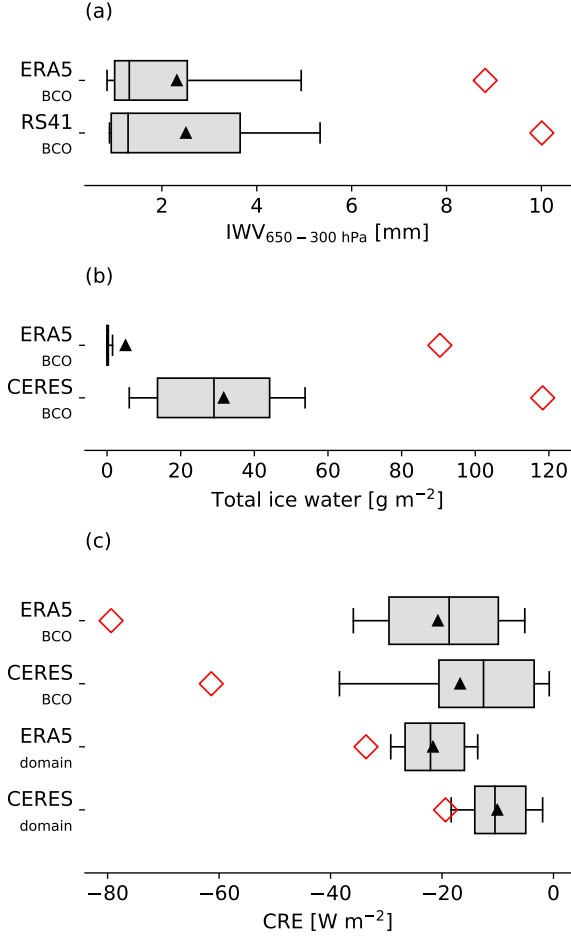

**Figure 13.** Boxplots of **(a)** daily mean $IWV_{650\text{-}300\,hPa}$ from ERA5 and the atmospheric soundings (RS41) at the BCO, **(b)** daily mean total ice water from ERA5 and CERES (note that many data points are based on spatial interpolation due to missing data, see Fig. 3) at the BCO, and **(c)** daily mean CRE from ERA5 and CERES at the BCO and as area-weighted mean over the domain 10-20° N, 50-60° W (red box in Fig. 5a; ERA5$_{domain}$ and CERES$_{domain}$). The boxplots (25, 75 percentiles as box; 10, 90 percentiles as whiskers; median as line in the box, mean as black triangle, outliers not shown) show values of 36 days (16 January to 20 February 2020). The values on 14 February 2020 are shown as red rhombi.

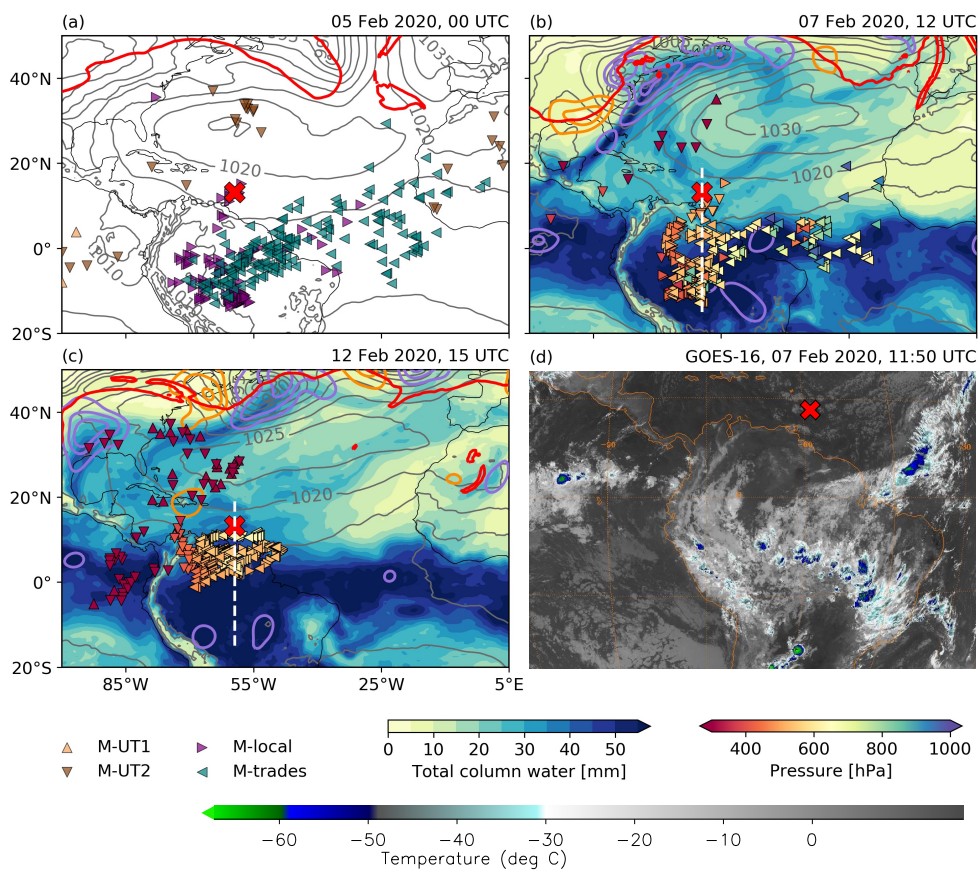

**Figure 14.** Synoptic situation over the North Atlantic and the position of backward trajectories from the BCO started in the layer 650-300 hPa on 14 February 2020 (00-21 UTC, every 3 h) at **(a)** 00 UTC 5 February, **(b)** 12 UTC 7 February, **(c)** 15 UTC 12 February 2020, and **(d)** cloud top temperature detected by the infrared channel ($10.2 - 11.6\,\mu$m) of the GOES-16 satellite at 11:50 UTC 7 February 2020 (note the different projection). Trajectory positions are shown as triangles coloured according to the **(a)** airstream or **(b,c)** pressure. Color shading shows total column water, and contours show sea level pressure (grey; 5 hPa intervals), upward and downward winds at 500 hPa (purple and orange, respectively; $\pm0.2, \pm0.4, \pm0.6\,\mathrm{Pa\,s}^{-1}$), and 2 pvu at 320 K (red). The red cross marks the location of Barbados and the white dashed line the location of the cross section shown in Fig. 16. A Gaussian filter was applied to the vertical winds for better readability. Data: **(a)**-**(c)** ERA5, **(d)** GOES-16.

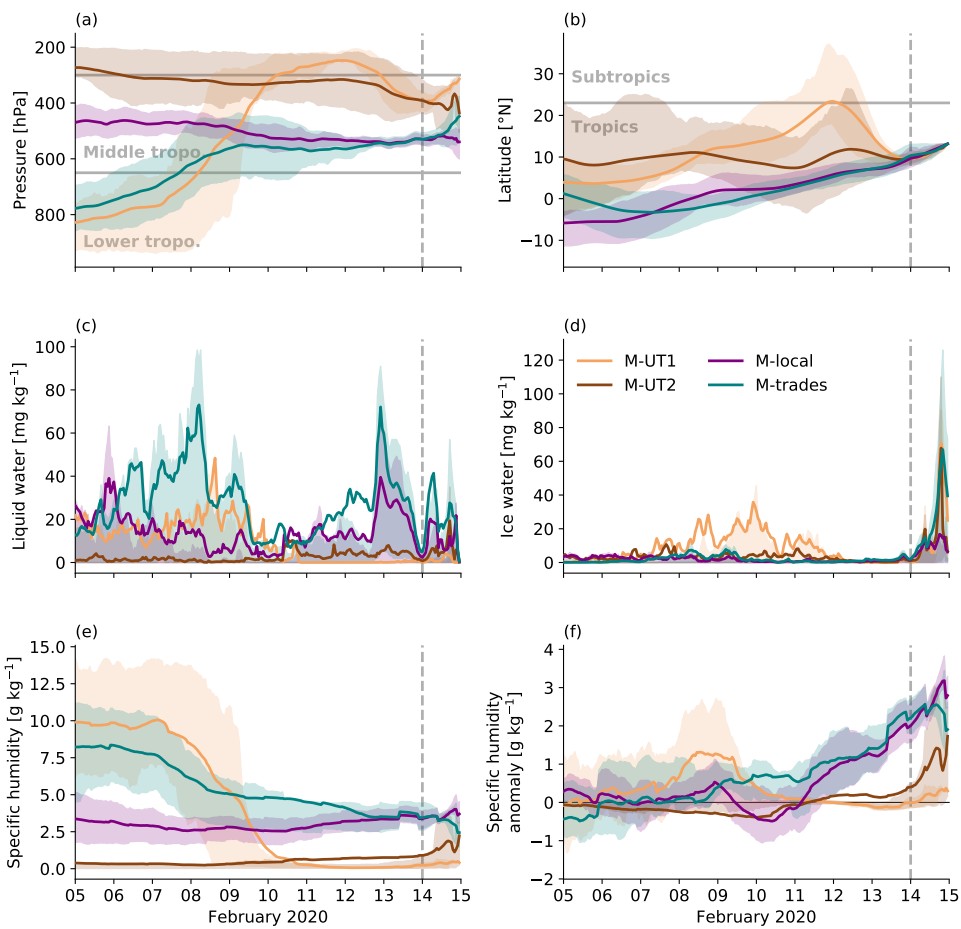

**Figure 15.** Time series along the four airstreams arriving at the BCO in the layer 650-300 hPa on 14 February 2020. The variables shown are **(a)** pressure, **(b)** latitude, **(c)** liquid water content, **(d)** ice water content, **(e)** specific humidity, and **(f)** anomaly in specific humidity calculated relative to the level-specific campaign mean value at the location of the trajectory. Thick lines show mean values and the shading the 25-75-percentile range. The number of trajectories assigned to each airstream is given in Table 4. Note that from 00 UTC 14 February 2020 (gray dashed line) onward the number of trajectories in the airstreams continuously decreases as they reach the BCO. Data: ERA5.

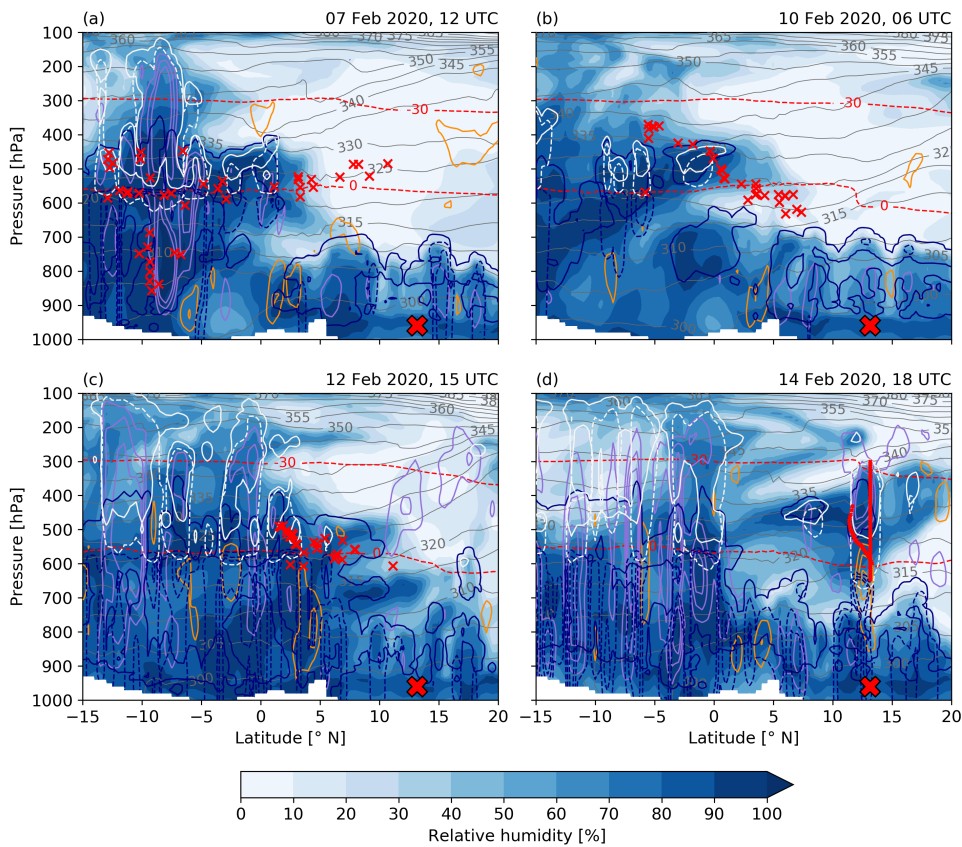

**Figure 16.** Cross section along -59.43° W from 15° S to 20° N (white dashed line in Fig. 14) and the position of backward trajectories from the BCO started in the layer 650-300 hPa on 14 February (00-21 UTC, every 3 h) at **(a)** 12 UTC 7 February, **(b)** 6 UTC 10 February, **(c)** 15 UTC 12 February, and **(d)** 18 UTC 14 February 2020. Trajectory positions not more than $\pm 1°$ W away from the cross section are shown as red thin crosses, which are miniaturized in **(d)** for better readability. Color shading shows relative humidity, and contours show potential temperature (grey; 5 K intervals), 0 and $-30°$ C-isotherm (red dashed), upward and downward winds (purple and orange, respectively; $\pm 0.2$, $\pm 0.4$, $\pm 0.6$ Pa s$^{-1}$), liquid (blue continuous), snow (white dashed), ice (white continuous; 10 mg kg$^{-1}$), and rain water (blue dashed; 1, 5, 10 mg kg$^{-1}$). The red thick cross marks the location of Barbados. Data: ERA5.

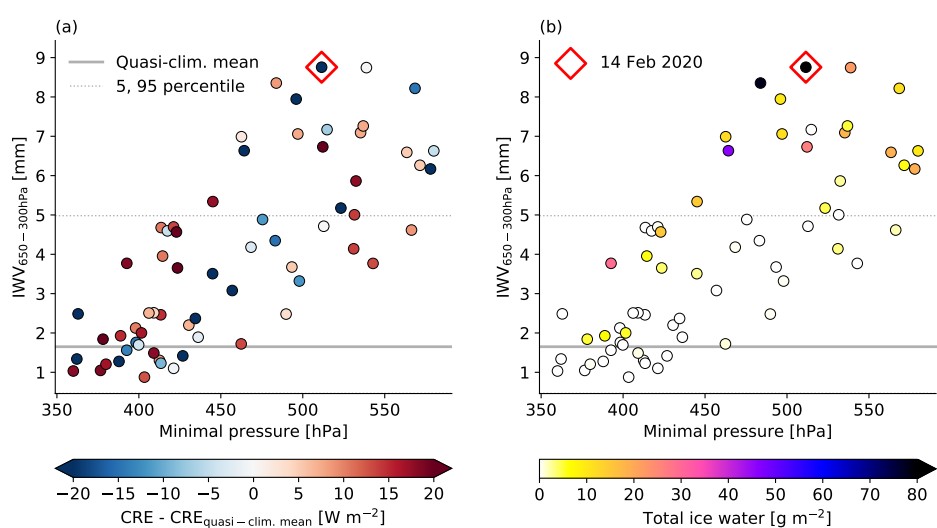

**Figure 17.** The relation between daily mean $IWV_{650\text{-}300\,hPa}$, the mean of the minimal pressure reached during the ten-day journey by the trajectories fulfilling the tropical detrainment criteria and **(a)** the anomaly in daily mean CRE relative to the quasi-climatological mean or **(b)** daily mean total ice water at the BCO is shown for the 29 TUD and TMD days. The quasi-climatological means are shown as gray continuous lines, and the 5 and 95 percentile as gray dashed lines. The red rhombus indicates 14 February 2020. Note that the daily mean values are based on three hourly data, thus the values are not directly comparable to Fig. 13. Data: ERA5.

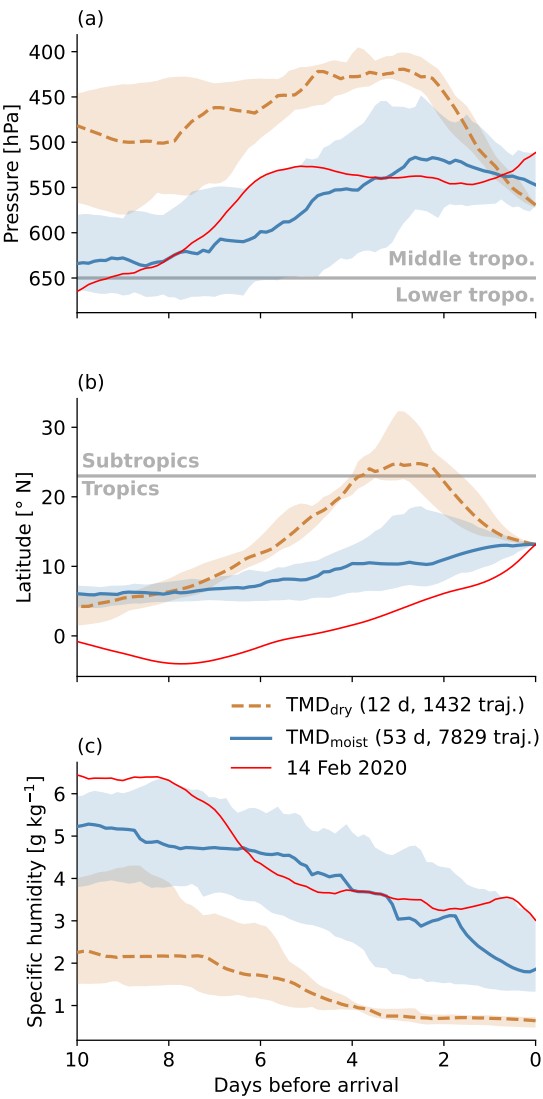

**Figure 18.** Time series along the TMD trajectories arriving at the BCO in the layer 650-300 hPa on the 12 TMD days with negative (brown dashed) and 53 TMD days with positive (blue continuous) anomalies in $\text{IWV}_{650\text{-}300\,\text{hPa}}$ during the quasi-climatological period. The variables shown are **(a)** pressure, **(b)** latitude, and **(c)** specific humidity. Thick lines show the median value, and the shading the 25-75-percentile range. The mean of the TMD trajectories of 14 February 2020 is shown as red thin line. Data: ERA5.

**Table 1.** Size and humidity of the five airstreams arriving above the BCO in the layer 1000-650 hPa on 22 January 2020. Data: ERA5.

| Airstream | Number of trajectories | | Mean specific humidity per air parcel | Contribution to IWV$_{1000\text{-}650\,\text{hPa}}$ |
|---|---|---|---|---|
| | Absolute count | Fraction of 1200 | | |
| L-EDIwcb | 78 | 7 % | 11.7 g kg$^{-1}$ | 8 % |
| L-EDIs | 311 | 26 % | 12.5 g kg$^{-1}$ | 34 % |
| L-EDIw | 143 | 12 % | 12.9 g kg$^{-1}$ | 16 % |
| L-MT | 423 | 35 % | 4.5 g kg$^{-1}$ | 16 % |
| L-LT | 245 | 20 % | 12.3 g kg$^{-1}$ | 26 % |

**Table 2.** The 44 EDI days from the quasi-climatological period shown in bold/cursive print if belonging to the five EDI days with the lowest/highest daily mean low-level divergence at the BCO. 2012 and 2015 are not shown because they have no EDI days. Data: ERA5.

| Year | 2010 | 2011 | 2013 | 2014 | 2016 | 2017 | 2018 | 2019 | 2020 |
|------|------|------|------|------|------|------|------|------|------|
| Date | **27**-28 **Jan** | 1 Jan<br>18 Jan<br>31 Jan<br>16 Feb<br>**26 Feb**<br>28 Feb | 14 Jan<br>*17 Jan*<br>*2,3,4 Feb* | 4 Jan<br>14-*17 Jan*<br>27-28 Jan<br>15 Feb<br>18-19 Feb | 13-*16 Jan* | 1 Jan<br>**13 Jan** | 31 Jan-1 Feb<br>23-*25 Feb*<br>28 Feb | 1-2 Jan<br>**27 Jan**<br>23-25 Feb | **22**-24 **Jan** |
| Count | 2 | 6 | 5 | 10 | 4 | 2 | 6 | 6 | 3 |

**Table 3.** Pearson correlation coefficients matrix for the daily mean $IWV_{1000\text{-}650\,hPa}$, the daily precipitation totals, the daily mean CRE and the daily mean divergence at $950\,hPa$ at the BCO for the 44 EDI (608 nonEDI) days of the quasi-climatological period. Data: ERA5.

| | Precipitation [mm] | CRE [$W\,m^{-2}$] | Divergence [$10^{-6}\,s^{-1}$] |
|---|---|---|---|
| $IWV_{1000\text{-}650\,hPa}$ [mm] | 0.71 (0.53) | -0.22 (-0.28) | -0.66 (-0.52) |
| Precipitation [mm] | | -0.35 (-0.28) | -0.84 (-0.43) |
| CRE [$W\,m^{-2}$] | | | 0.37 (0.26) |

**Table 4.** Size and humidity of the four airstreams arriving above the BCO in the layer 650-300 hPa on 14 February 2020. Data: ERA5.

| Airstream | Number of trajectories | | Mean specific humidity | Contribution to |
| | Absolute count | Fraction of 1200 | per air parcel | IWV$_{650\text{-}300\,\text{hPa}}$ |
| --- | --- | --- | --- | --- |
| M-UT1 | 84 | 7 % | 0.4 g kg$^{-1}$ | 1 % |
| M-UT2 | 205 | 17 % | 0.8 g kg$^{-1}$ | 6 % |
| M-local | 316 | 26 % | 3.1 g kg$^{-1}$ | 33 % |
| M-trades | 595 | 50 % | 3 g kg$^{-1}$ | 60 % |

**Table 5.** The 65 TMD days from the quasi-climatological period. Data: ERA5.

| Year | 2010 | 2011 | 2012 | 2013 | 2014 | 2015 | 2016 | 2017 | 2018 | 2019 | 2020 |
|---|---|---|---|---|---|---|---|---|---|---|---|
| Date | 11-13 Feb<br>23-28 Feb | 7-8 Jan<br>21 Jan<br>23-25 Feb | 24-25 Jan<br>8 Feb<br>12 Feb | 5 Jan<br>19 Jan | 7-10 Feb | 15-16 Jan<br>18-20 Jan<br>28-29 Jan<br>5-8 Feb<br>12-14 Feb | 4 Jan<br>28 Jan<br>5-6 Feb | 29-30 Jan<br>3 Feb<br>6 Feb<br>25-27 Feb | 11 Jan<br>14 Jan<br>19 Feb | 7-8 Feb | 5 Jan<br>15 Jan<br>7-8 Feb<br>14-19 Feb |
| Count | 9 | 6 | 4 | 2 | 4 | 14 | 4 | 7 | 3 | 2 | 10 |

**Table 6.** Pearson correlation coefficients matrix for the daily mean $IWV_{650\text{-}300\,hPa}$, total ice water, CRE at the BCO and the minimal pressure attained during the ten-day journey by the trajectories meeting the TMD criterion for the 65 TMDs (and the remaining 587) days of the quasi-climatological period. Data: ERA5.

| | Total ice water [$\mathrm{g\,m^{-2}}$] | CRE [$\mathrm{W\,m^{-2}}$] | Minimal pressure [hPa] |
|---|---|---|---|
| $IWV_{650\text{-}300\,hPa}$ [mm] | 0.59 (0.36) | -0.17 (-0.16) | 0.76 |
| Total ice water [$\mathrm{g\,m^{-2}}$] | | -0.22 (0.01) | 0.26 |
| CRE [$\mathrm{W\,m^{-2}}$] | | | -0.17 |