# Peer review of "Lagrangian formation pathways of moist anomalies in the trade-wind region during the dry season: two case studies from EUREC4A"

_Weather and Climate Dynamics, 2021_

## Referee Comment (RC2)

Review of: Lagrangian formation pathways of moist anomalies in the trade-wind region during the dry season: two case studies from EUREC4A
authors: Leonie Villiger, Heini Wernli, Maxi Boettcher, Martin Hagen, and Franziska Aemisegger
Manuscript #: 10.5194/wcd-2021-42

recommendation: minor/major revision

This is a thorough study of two types of moist anomaly intrusions seen during the EUREC4A campaign, with the effort made to assess their climatological representativeness. The work is novel. I enjoyed reading this, and appreciate the effort the authors made to provide a polished document that could be easily reviewed along its way to publication.

My main comment is that I wished for more discussion on how well ERA5 can be used to assess the atmospheric aspects the authors most drew on. We are told on page 5 that the ERA5 variables of most import are IWV, precipitation, and CRE. There is no discussion here on their representativeness. Comparisons of ERA5-IWV to the radiosonde-derived values at BCO are shown in Fig. 6, along with precipitation. The IWV comparison clearly indicates moister radiosondes, most likely in the boundary layer (based on fig. 5b and fig. 12b). How about showing IWV comparisons for multiple layers? And why no comparison to CERES-derived CRE values? How much faith should we have in the ERA5 CRE values given the large spread in values reported by the authors - could any of this reflect a systematic over/underestimation by ERA5 of low/mid/high clouds? There is some discussion on p. 8 on how the ERA5 CRE compare to the satellite-derived values in the Bony 2020 study, but to place that discussion there feels adhoc. Better is to devote a section to a more cohesive discussion of the ERA5 data strengths/weaknesses. Is there any literature to draw on that has compared ERA5 variables to data (e.g. AIRS?)? In addition, further along in the manuscript, we see the ERA5 humidity variables, evaporation and liquid water along the trajectories. Why are these not listed on p.5? Surely there is more the authors can say about the strengths and weaknesses of the ERA5 physical variables for their study. And if not, the authors need to at least mention this shortcoming of their analysis.

A further comment is, in attempting to reconcile how an EDI can suppress/encourage shallow convection, it seems to me that there must be some modulation of the temperature profile that either encourages or discourages a (dis)stabilization of the atmosphere. Or is this not the case because the temperature profile equilibrates so quickly in the tropics? I would appreciate seeing some analysis/discussion of the moist anomaly intrusions impacts on the temperature profiles somewhere, primarily for the EDI case as the impact of the shelf clouds detrained as part of the TMD case is easier to understand.

Minor/specific comments:

Abstract: line 10: the abstract jumps into describing the low-level moist anomaly here. Would suggest labeling this as 'case 1' to distinguish this one from the other.
Line11: "Its" I believe would refer to the negative CRE of the previous sentence, as written. Is this the intent?
Line 21: "of the long-range transport" -> "of long-range moisture transport"

p. 3, lines 78-79: I don't quite follow how an EDI is already interacting with PBL air on day 2, is this implying the trajectory is starting at 500-600 hPa? (since they tend to descent 400 hPa in 2 days?)

p. 6, section 2.2: I presume the Lagranto trajectories rely on the ERA5 subsidence velocities. Have these been compared to the EUREC4A dropsonde-circle-derived vertical velocities - is there any assessment known of the ERA5 values?

p. 7 section 4.1: The 'Fish' cloud structure described here, similar to others, is bordered by cloud-free regions, suggesting (to me) an enhancement to the subsidence at the mesoscale. Do the authors see cloud features like this in the ERA5 data as well? Would a spatial plot of ERA5 liquid water path provide a useful comparison to the MODIS image shown in FIg. 5?

P. 12 bottom - p. 13 top: this discussion on future work seems a better fit for the end of the manuscript.

P. 14, line 434: the references provided are all to low-level mixed-phase clouds I believe, for which the shortwave CRE will be more dominant. If the authors are able to draw in literature more pertinent to the mid-level mixed-phase cloud CRE that would be more pertinent. I provide some suggestions below but am not sure if those also  evaluate the CRE.

Figures:
Fig 2: I believe this is entirely ERA5, regardless it would be useful for the figure caption to clarify.

FIg 3: I do not see the red shadings.

Other potentially useful references:
Casey, S. P. F., Dessler, A. E., & Schumacher, C. (2009). Five-Year Climatology of Midtroposphere Dry Air Layers in Warm Tropical Ocean Regions as Viewed by AIRS/Aqua, *Journal of Applied Meteorology and Climatology*, *48*(9), 1831-1842.
Is their data description consistent with your ERA5 inferences?

Zuidema, P., B. Mapes, J. Lin, C. Fairall, and G. Wick, 2006: The Interaction of Clouds and Dry Air in the Eastern Tropical Pacific. *J. Climate*, **19**, no. 18, pp. 4531-4544.
Their discussion of how the latent cooling from evaporating falling ice particles helps maintain a stability structure that can help sustain the upper-level stratiform cloud deck seems relevant to the TMD discussion on p.13.

Relevant to the radiative impact of mid-level mixed-phase clouds:
Bourgeois, Q., Ekman, A. M. L. L., Igel, M. R., and Krejci, R.: Ubiquity and impact of thin mid-level clouds in the tropics, Nat. Commun., 7, 12432, https://doi.org/10.1038/ncomms12432, 2016.
Sassen, K. and Wang, Z.: The Clouds of the Middle Troposphere: Composition, Radiative Impact, and Global Distribution, Surv. Geophys., 33, 677–691, https://doi.org/10.1007/s10712-011-9163-x, 2012.
Adebiyi, A. A., P. Zuidema, I. Chang, S. P Burton and B. Cairns, 2020: Mid-level clouds are frequent above the southeast Atlantic stratocumulus clouds. *Atmos. Chem. Phys.*, **20**, p. 11025-11043, 10.5194/acp-20-11025-2020

---

## Author Response (AR1)

**Authors' reply to reviewer comments of the manuscript**: "Lagrangian formation pathways of moist anomalies in the trade-wind region during the dry season: two case studies from EUREC$^4$A" by Leonie Villiger, Heini Wernli, Maxi Boettcher, Martin Hagen, and Franziska Aemisegger

We thank both reviewers for their insightful comments, which we address in detail in our responses below. They helped to further improve the presentation of our results and led to the following main changes in the revised manuscript:

1) The criteria for the identification of TMDs in the quasi-climatological period has been generalized and the related sections have been updated accordingly.
2) More evidence regarding the position of EDI trajectories relative to the frontal surface is now provided in Supplement 2 and the similarity of the investigated cases to kata- and anafronts is discussed.
3) A new subsection (2.3) addressing ERA5's representativeness in the vicinity of Barbados during EUREC4A has been added to the manuscript, referring to new figures in Supplement 1.

Reviewer's comment, *Authors' reply (line numbers refer to the revised version of the manuscript including track changes)*
* * *
1. **Reviewer #1:** Quasi-climatological identification of TMDs: the motivation behind finding a similar pathway to the rather peculiar case is not clear. In principle, there can be detrainment from cumulonimbus anywhere in the tropics, not necessarily in the southern hemisphere, and indeed many of the M-local and M-trades of the February 2020 case trajectories do not originate from the southern hemisphere. Please explain why you focus on this specific pathway (which is indeed very specific, as the ice content value shows), or alternatively generalize the tropical detrainment criteria. I actually find the results in section 5.3 to be somewhat trivial, as they result directly from the definition of TMD days by their trajectories (e.g., pressure and latitude evolution, relation between minimum pressure and IWV). Also given the less robust relation to CRE, and the already many figures, I suggest to skip this subsection along with Figs. 17 and 18, and summarize the quasi-climatological context with less detail and focus on the mid-level anomaly only.

   *We agree that the definition of TMD trajectories as trajectories that originate from the Southern Hemisphere is somewhat arbitrary. Therefore, we have generalized the TMD criteria and slightly shortened section 5.3. However, we refrain from merging section 5.3 with 5.2 because we want to keep the structural symmetry of case study 1 (EDI) and 2 (TMD). We realize that the paper is rather long and has many figures, but we justify this with the fact that the paper addresses two topics (EDI, TMD). The reasoning behind the current structure (with the many subsections) is that it allows easy selective reading, depending on the readers interests. In detail we have made the following adjustments (see revised manuscript for actual content of the adjustments):*
   - *Sect. 5.3 has been adapted, starting with the new, generalized TMD selection criteria (we no longer differentiate between mid- and upper-level detrainment)*
   - *Sec. 5.4, paragraphs addressing TMDs have been updated*
   - *Fig. 17 has been replaced with new version*
   - *Fig. 18 has been updated and shortened by one panel (another reason to keep Fig. 18 is given in item 19 of this document)*
   - *Table 5 has been updated with new values*
   - *Table 6 has been updated with new values*
   - *Supplement 1: new Fig. S1.2 has been added, showing the effect of different TMD (and EDI, see item 8 of this document) selection criteria. The former Fig. S1.2 has been removed because we add some more Supplement figures in the context of this reply and believe that the former Fig. S1.2 is less relevant.*

   *As can be seen from Fig. 17 several TMD cases in the climatology are related to elevated total ice content over Barbados. The CRE is indeed unrelated to IWV or the ice water content. This observation agrees with existing literature showing that the CRE of mid-level clouds depends on many factors (that we do not investigate explicitly), such as the ice-to-liquid ratio (Sassen and Wang, 2012), the thickness of the cloud, the underlying surface, additional cloud layers at other levels (Bourgeois et al., 2016), and not only on IWV or the ice water content. Another factor is the definition of the 24-h averaging window. We have seen (see e.g., Fig. 3 in the paper or*

*Supplement 1 new Fig. S1.7-S1.8) that mid-level clouds cause very strong responses in the CRE of opposed sign during day- and night-time. Thus, shifting the 24-h averaging window will change the daily mean CRE. For future studies we recommend testing different, e.g., longer averaging windows (on the synoptic time scale).*

2. **Reviewer #1:** The analysis concerning EDIcon/div shows interesting and coherent results regarding the precipitation response to EDI events. However, it raises some doubts regarding the position of the EDI trajectories with respect to the slanted frontal surface. Since the front is identified on the 850-hPa surface and divergence is considered at 950 hPa, it is possible that the EDI airmasses seem to reach ahead of the front to its warm side, but are in fact still at the cold sector since they are located closer to the surface. For example, the bottom panels of S2.4 suggest that there is divergence directly below the warm side of the front over Barbados, where the trajectories reach (or is this partly hidden by the large marker?). It is clear that the con/div conditions prevail locally at the BCO and those are directly related to the IWV and precipitation there. However, the statements about the position of the EDI trajectories with regards to the front need more evidence. As the mesoscale variations of the divergence field near the frontal region can have sharp variations, there is more direct evidence needed to state that the EDIcon trajectories indeed reach the warm side of the front by the ageostrophic circulation. Furthermore, if indeed the EDI trajectories – front location be substantiated, can the discussion on the EDI-front interaction be further related to existing knowledge on kata/ana fronts and precipitation in the midlatitudes?

   *Thank you, we address this important comment in detail, considering four complementary aspects:*

   a) ***Motivation for showing equivalent potential temperature on 850 hPa and divergence on 950 hPa****: We show $\theta_e$ on 850 hPa because (1) it is common practice to identify surface fronts on this pressure level (e.g., Schemm et al., 2015) to avoid the influence of the turbulent boundary layer, and (2) because (even though the strength would decrease) the latitudinal location of the horizontal $\theta_e$ gradient would not change much if it were identified on 950 hPa (Fig. R1b). Due to very strong diabatic modification of the boundary layer air, two cold fronts would be detected at 950 hPa (at 17°N and 25°N). The motivation to show divergence on 950 hPa is (1) to connect to existing literature (i.e., Schulz et al., 2021, who linked 950 hPa divergence to the mesoscale cloud organisation pattern Fish), and (2) because we are interested in the surface weather in Barbados and therefore want to describe near-surface dynamics. Indeed, the discussed vertical motion is very shallow (only reaching ~750 hPa; see vertical dipoles in divergence field above Barbados in Fig. R1d), causing a weakened or even reversed horizontal divergence signal on ~850 hPa. Thus, only by showing divergence at p > ~900 hPa, we can capture relevant near surface dynamics.*

[Figure]

***Fig.** R1: Synoptic situation over the North Atlantic at 07 UTC on 22 January 2020 in the (a,c) horizontal and (b,d) vertical dimension. Shown are (a) equivalent potential temperature at 850 hPa and (c) divergence at 950 hPa (shading), sea level pressure (grey; 5 hPa intervals), cyclone masks (black), 2 pvu at 320 K (red/pink), precipitation (blue; 0.5 mm h⁻¹), the location of the cross section displayed on the right (black straight line); (b) equivalent potential temperature and (d) divergence (shading), potential temperature (grey; 1 K intervals), rain water (blue; 5, 10, 20 mg kg⁻¹), the location (inside the black dashed box) of the backward trajectories started in the layer 1000-650 hPa on 22 January 2020 (09-15 UTC, every hour). The BCO is indicated by the red cross.*

b) ***Location of air parcels relative to slanted frontal surface/Evidence that air parcels move from cold to warm side of the front:** To show the location of the EDI air parcels relative to the slanted frontal surface, we replaced two figures in Supplement 2 and added two new ones. The adapted/new figures in Supplement 2 show horizontal and vertical cross sections (oriented along the EDI air parcels) through the three-dimensional θₑ and divergence fields for the two exemplary, contrasting cases (22 January 2020, and 15 January 2014). The figures show that the EDI air parcels in the 22 January 2020 case indeed cross the front and arrive on the front's warm side (Fig. S2.4, S2.5). In contrast, in the 15 January 2014 case (Fig. S2.6, S2.7), the EDI air parcels remain behind the front and arrive in Barbados together with the cold front's divergent cold side. The new figures also emphasize the absence of baroclinicity above the boundary layer in Barbados (a key characteristics of extratropical fronts). Thus, we decided to better emphasise the dynamical difference between extratropical cold fronts and trailing cold fronts in the subtropics. For this reason, we added a paragraph in the introduction. More specifically, the following adaptions have been made:*

- *Abstract: changed "cold front" to "trailing cold front"*
- *L96-L102: added new paragraph on the difference between extratropical baroclinic fronts and cold fronts in the subtropics (see revised manuscript for content)*
- *L414, L422, L425, L428: added references to Supplement 2*
- *L455-L458: adapted text (new text underscored): "[...] the EDI overtakes the cold front by entering the boundary layer on the cold front's cold side and subsequently ascending on the font's warm side. In the second case, the front and the EDI are spatially and temporally separated, with the front propagating faster than the EDI and the EDI entering the boundary layer only when arriving in Barbados."*
- *Supplement 2: added new figures S2.5-S2.7 (Fig. S2.6 became S2.8) and adapted text*

c) ***Relation to kata/ana front****s: The following statements are based on the new figures S2.4-S2.7 in Supplement 2.The considered cases show that the trailing cold fronts in the subtropics adopt characteristics similar to the ones of katafronts: (1) The upgliding of warm and moist air masses along the backward tilted frontal surface (associated with precipitation falling into the cold sector) is not observed, presumably due to the descending EDI air masses aloft. (2) Instead, we find a narrow precipitation band along the surface cold front. (3) Additionally, the vertical growth of the convection at the surface cold front is capped by air with low equivalent potential temperature, which might be part of the EDI (however, a backward tracing of the air parcels would be necessary to substantiate this statement). The following adjustments have been made (see revised manuscript for content):*
    - *L103-L117: added new paragraph introducing ana- and katafronts*
    - *L464-L468: added new paragraph summarizing observed katafront characteristics in analysed cases*
d) ***(De)Stabilisation of the atmosphere through the impact of the EDI*** *(see also item 12 of this document): The new cross sections in Supplement 2 (Fig. S2.4-S2.7) illustrate that the EDI stabilizes the cold sector boundary layer top. We assume that this is due to the adiabatic compression of the descending EDI air in the free troposphere and simultaneously the longwave cooling of the top of the boundary layer (which is enhanced due to the dry, clear-sky free troposphere promoted by the arrival of the EDI). In the boundary layer at the trailing cold front and behind, we hypothesize that the EDI, if it enters the boundary layer, promotes turbulence through an enhancement of surface sensible and latent heat fluxes (due to the strong vertical temperature and humidity gradients created by the EDI).*

**Reviewer #1:** Line 201-202: it is not immediately clear which balance is referred to here. Please explain. *Here we refer to the balance between adiabatic compression and radiative cooling from Salathé and Hartmann (1997), i.e. 35 hPa (24 h)$^{-1}$ corresponding to about 1 hPa h$^{-1}$), which is the expected climatological subsidence in the subtropics (see also Holton and Hakim, 2013, Chapter 11).*
- *L279, added new text (underscored): "[…] corresponding to the expected values (of roughly 35 hPa (24 h)$^{-1}$, see Salathé and Hartmann, 1997; Holton and Hakim, 2013) from the balance between adiabatic compression and radiative cooling […]"*

3. **Reviewer #1:** Line 205: is the inversion represented also in the temperature profile? *Yes, the inversion is also present in the temperature profile, but its height depends on the criteria for its identification (i.e. the threshold of the required temperature increase). We have tested several thresholds for the vertical temperature profiles from the BCO sounding and found that 0.4°C over the depth of a given layer with continuous temperature increase captures the inversion reasonably well. Two exemplary soundings are shown in Fig. R2. In the manuscript we have made the following adjustments:*
   - *Fig. 2 and 4: The inversion height derived from the BCO soundings is now displayed and the caption, in which the identification method is shortly summarised, were adapted.*
   - *L284, text adaptation (new text underscored): "[…] and a strong inversion associated with very dry conditions (at 600 to 900 hPa; Fig. 2 and 4).*

[Figure]

*Fig. R2:* Measurements from the radio sounding launched at the BCO at (a) 14:44 UTC on 22 January, and (b) 10:42 UTC on 14 February 2020. Shown are dewpoint (left black line) and temperature (right black line), the lifting condensation level (blue line) derived with the method from Romps (2017), the lowest temperature inversion (red line) with a minimal temperature increase of 0.4°C over the depth of the inversion layer, and dry/humid layers (brown/blue shading) where the smoothed relative humidity profile decreases to below 20%/increases to above 80%, respectively.

4. **Reviewer #1:** 8d and accompanying text: it may not be clear that here you refer to surface evaporation since you also mention the evaporation of hydrometeors into the dry airmass. Please clarify this in the text and caption. *We have adapted the figure y-axis label to "Surface evaporation" and also adapted the caption text to "surface evaporation". In the text we have made sure that evaporation is always specified as "surface/ocean evaporation" or "evaporation of hydrometeors".*

5. **Reviewer #1:** What is the significance of L-DIwcb? Is it needed to separate the analysis? *Whether L-DIwcb is investigated as individual airstream or as part of another L-DI\* airstream doesn't matter for the results. However, for the authors it came as a surprise that the air parcels, which are first part of a warm conveyor belt (WCB) immediately move into a dry intrusion airstream after a short passage through the jet stream. With the L-DIwcb we merely wanted to illustrate the impressive vertical displacement within a short time interval of individual airstreams along a path with rapid ascent followed by rapid descent, which (we assume) many readers are not aware of. This implies a very fast moisture turnover along the large-scale flow. Furthermore, the three DI airstreams are associated with contrasting histories before initiating their descent, a fact that we wanted to highlight. Therefore, we would like to keep the airstream definition the way it is, even if it is not crucial for our findings.*

6. **Reviewer #1:** Line 275: actually, the specific humidity increases to easily above 10 g/kg, roughly double the typical North Atlantic values (~6 g/kg). Is this due to the relatively high temperature in the tropical region and thus higher specific humidity at saturation? *Yes, exactly. We expect "above typical North Atlantic values" in tropical regions of the North Atlantic due to the higher specific humidity at saturation (as stated by the reviewer). However, we assume that for the creation of the observed local moist anomaly (anomalous relative to tropical mean conditions; Fig. R3h), other processes (besides higher temperature) are needed. When the trajectories descend (Fig. R3b) into the tropics (Fig. R3a), their temperature (Fig. R3d) gradually increases. Specific humidity is more strongly influenced by the vertical location (i.e., above, in, below clouds) of the air parcel than by its latitudinal position or temperature.*

[Figure]

**Fig. R3**: *Evolution along the backward trajectories started from the BCO in the layer 1000-650 hPa on 22 January (00-23 UTC, every hour) of (a) latitude, (b) pressure, (c) surface evaporation, (d) temperature, (e) liquid water, (f) specific humidity, (g) rain water, and (h) anomaly of specific humidity relative to the campaign mean at the current location of the air parcel. Shown are the mean and 25-75 percentile range for the five airstreams (see legend in h) defined in the manuscript.*

7. **Reviewer #1:** 4.3: what is the sensitivity to the 5% EDI criterion? Does the event during 8-9 February 2020 qualify as an EDI event using a lower percentage or a different time span? *Yes, it does. If the percentage were reduced to 4%, 7 February 2020 would qualify as an EDI event. If the time span were expanded from 4 to 6 days before arrival, 7-9 February (with 7-10% EDI trajectories) would qualify as EDI. We have adapted the following:*
   - *Supplement 1, new Fig. S1.3 addressing the sensitivity of the EDI (and TMD) selection criteria*
   - *Footnote on p. 13: added a reference to the new Fig. S1.3 in Supplement 1.*
   - *L660-L661: mentioned sensitivity of EDI frequency to selection criteria*
   - *L695-L696: mentioned sensitivity of TMD frequency to selection criteria*

8. **Reviewer #1:** How well is the vertical velocity represented at the BCO region in ERA5 compared to observations from BCO? This is especially important under convective conditions e.g. in Fig.

16d. *We address this comment together with a comment from reviewer #2. See our response in item 11 of this document.*

9. **Reviewer #1: Technical corrections:**
   - Line 273: delete "in". *"setting in" has been replaced with "starting"*
   - S2.4 caption first line: delete "on". *"on" has been deleted*
   - Line 319: add "y" to "dail" *"y" has been added*
   - 9 and 17 captions: replace "geographical" by "geometrical" both captions have been adapted
   - 14d: mark the location of BCO, as the domain is shifted compared to the other panels. *marker has been added*
   - 17: correct the pressure levels of the IWV on the y axes. *y axes have been adapted*
   - Line 489: replace "to" with "from". *"to" has been replaced with "from"*
* * *
10. **Reviewer #2:** My main comment is that I wished for more discussion on how well ERA5 can be used to assess the atmospheric aspects the authors most drew on. We are told on page 5 that the ERA5 variables of most import are IWV, precipitation, and CRE. There is no discussion here on their representativeness. Comparisons of ERA5-IWV to the radiosonde-derived values at BCO are shown in Fig. 6, along with precipitation. The IWV comparison clearly indicates moister radiosondes, most likely in the boundary layer (based on fig. 5b and fig. 12b). How about showing IWV comparisons for multiple layers? And why no comparison to CERES-derived CRE values? How much faith should we have in the ERA5 CRE values given the large spread in values reported by the authors - could any of this reflect a systematic over/underestimation by ERA5 of low/mid/high clouds? There is some discussion on p. 8 on how the ERA5 CRE compare to the satellite-derived values in the Bony 2020 study, but to place that discussion there feels adhoc. Better is to devote a section to a more cohesive discussion of the ERA5 data strengths/weaknesses. Is there any literature to draw on that has compared ERA5 variables to data (e.g. AIRS?)? In addition, further along in the manuscript, we see the ERA5 humidity variables, evaporation and liquid water along the trajectories. Why are these not listed on p.5? Surely there is more the authors can say about the strengths and weaknesses of the ERA5 physical variables for their study. And if not, the authors need to at least mention this shortcoming of their analysis.

*Thank you, we address this important comment in detail, with the following points:*

a) ***Representativeness of ERA5***: *As the aim of the study is not primarily to evaluate the (rather new) reanalysis ERA5, we only provide punctual comparisons between ERA5 and observational data sets. In response to the comment, we have included the satellite product CERES (variables CRE, TCIW) in our analysis (Figs. 3, 6, 13), added a new subsection (2.3) and several text fractions, and extended Supplement 1. The new subsection summarizes all ERA5-to-measurement comparisons, including IWV over the two layers discussed in the case studies, and addresses the systematic underestimation of mid- to high-level clouds in ERA5 during EUREC4A and a case study specific overestimation of low-level clouds. Please note that the spatial data coverage of CERES TCIW (not CRE) is low (Fig. R4, left) and often only interpolated data (Fig. R4, right) could be used for the comparison to ERA5.*

[Figure]

**Fig. R4**: *Exemplary time step illustrating (left) spatial coverage of CERES TCIW (named IWP on the files) and (right) spatially interpolated data to fill the gaps. The location of Barbados is marked by the red circle.*

*In detail, we made the following adjustments:*

- *Fig. 3: Added IWV from radiosonde measurements; added panel (d) showing total ice water from ERA5 and from CERES; added CERES observations in panel (e) showing CRE; adapted caption accordingly.*
- *Fig. 6: Added CERES CRE in panel (c); adapted caption accordingly.*
- *Fig. 13: Added CERES total ice water in panel (b) and CERES CRE in panel (c); removed the marker showing the values from 15 February 2020, as this comparison is no longer needed whit the new comparison to CERES; adapted caption accordingly.*
- *L143: added the following text: "[...] and different satellite products."*
- *L152-L154, new text added: "The variables CRE and total ice water are taken from the satellite product Clouds and the Earth's Radiant Energy System (CERES; NASA, 2017) distributed by the National Aeronautics and Space Administration (NASA). CERES data is available at an hourly temporal and a $1° \times 1°$ spatial resolution."*
- *L183-L185, removed the following text, as the information is mentioned in the newly added subsection 2.3: "Note that a direct comparison between ERA5 precipitation values and the measurements is difficult as the former represent averages over a model grid box while the latter yield information about the local conditions at the BCO."*
- *L191: added reference to new figure in Supplement 1*
- *L194, added text: "The CRE was also derived from the satellite product CERES (NASA, 2017)"*
- *L208-L247, added new subsection entitled "Representativeness of ERA5 in the vicinity of Barbados (see revised manuscript for content)*
- *L256, new text (underscored): "[...] about $-29$ to $-45$ W m$^{-2}$ compared to a campaign-mean value of $-21$ W m$^{-2}$ according to ERA5)."*
- *L260, new text added (underscored): "[...] -79 Wm$^{-2}$ in ERA5 and -61 Wm$^{-2}$ in CERES on 14 February, the most negative value during the campaign.*
- *L261-262, removed the following text: "On 15 February, however, the warming almost completely balanced the cooling such that the daily mean CRE matched the campaign mean value."*
- *L312, new text added (underscored): "of about $-38$ W m$^{-2}$ in ERA5 and $-47$ W m$^{-2}$ in CERES on 22 January 2020.*
- *L314, added reference to new figure in Supplement 1*
- *L315-L320, adapted existing and added new text (underscored): "[...] the CRE on 22 January 2020 reduces to values of about -27 W m$^{-2}$ in ERA5 and -8 W m$^{-2}$ in CERES, which is still anomalously negative compared to the rest of the campaign according to ERA5, but not according to CERES (Fig. 6c). Looking at the spatial distribution of the CRE in the considered domain (see exemplary time step in Fig. S1.7), we note that ERA5 overestimates the presence of liquid/low-level clouds compared to CERES, leading to a stronger cloud radiative cooling in ERA5. Generally, ERA5 shows 10 to 20 W m$^{-2}$ more negative net cooling over the $10°\times10°$ domain compared to CERES or the satellite-based study of Bony et al. (2020, their Fig. 5).*
- *L488-L490 and L517-L530: adapted existing text; removed content addressing 15 February, as this comparison is no longer of interest now that we have the comparison to CERES: "Above Barbados, the shelf cloud caused a positive anomaly in the daily mean of the IWV$_{650-300hPa}$ (Fig. 13a) and the TCIW (Fig. 13b) compared to the remaining campaign days. Similar to the first case study, the anomalies are stronger in the measurements than in the reanalysis. Both data sets indicate an enhanced cloud radiative cooling above Barbados and over the $10°\times10°$ domain (Fig. 13c). The absolute CRE values (at the BCO $-79$ W m$^{-2}$ in ERA5, and $-61$ W m$^{-2}$ in CERES; over the domain $-34$ W m$^{-2}$ in ERA5, and $-19$ W m$^{-2}$ in CERES) are lower in ERA5 than in CERES. This observation fits well with the higher TCIW recorded by CERES and the general overestimation of the cloud radiative cooling by ERA5, already observed in the first case study. In contrast to the first case study, however, the CRE anomaly is also negative over the domain (lowest two boxplots in Fig. 13c)."*
- *L713-L714: added data availability information of CERES*

> b) ***Listing of ERA5 humidity variables on p. 5:*** *The reasons why we don't list the ERA5 humidity variables (evaporation, liquid water) along the trajectories on p. 5 in section 2.1 (Characterisation of the local conditions in Barbados) are that (1) these are variables along the trajectories, therefore, they don't describe the local conditions, but the conditions during transport. (2) We only list the variables (i.e., IWV, precipitation totals, CRE), which we have computed from primary ERA5 variables (i.e., specific humidity, stratiform/convective precipitation, top thermal/solar radiation all/clear sky) on p. 5 in section 2.1. We don't list primary variables as we assume readers are familiar with ERA5 or (if not) refer to the provided literature (Hersbach et al., 2020). In response to the comment, we have adapted the manuscript and now list all ERA5 variables that we interpolated along the backward trajectories in Sect. 2.2 (Characterisation of the transport pathways towards Barbados).*
> - *L199-200, added new text: "[...] namely, specific humidity (absolute values and anomalies relative to the three-dimensional 3D campaign mean field), relative humidity, liquid/rain/snow/ice water content, and surface evaporation."*

11. **Reviewer #2:** A further comment is, in attempting to reconcile how an EDI can suppress/encourage shallow convection, it seems to me that there must be some modulation of the temperature profile that either encourages or discourages a (dis)stabilization of the atmosphere. Or is this not the case because the temperature profile equilibrates so quickly in the tropics? I would appreciate seeing some analysis/discussion of the moist anomaly intrusions impacts on the temperature profiles somewhere, primarily for the EDI case as the impact of the shelf clouds detrained as part of the TMD case is easier to understand. *We address this comment together with a comment from reviewer #1. See our response in item 2 of this document.*

12. **Reviewer #2:** Abstract:
    a) Line 10: the abstract jumps into describing the low-level moist anomaly here. Would suggest labeling this as 'case 1' to distinguish this one from the other. *We have adapted the sentence to (new text underscored): "The first case study about the low-level moist anomaly is characterised by an unusually thick cloud layer, high precipitation totals and a strongly negative CRE."*
    b) Line 11: "Its" I believe would refer to the negative CRE of the previous sentence, as written. Is this the intent? *No, the low-level moist anomaly was meant with "its". We have adapted the sentence to (new text underscored): "The formation of the low-level moist anomaly is connected to [...]". The same grammatical mistake was corrected for the second case study described in the abstract.*
    c) Line 21: "of the long-range transport" -> "of long-range moisture transport" *has been adapted*

13. **Reviewer #2**: p.3, lines 78-79: I don't quite follow how an EDI is already interacting with PBL air on day 2, is this implying the trajectory is starting at 500-600 hPa? (since they tend to descent 400 hPa in 2 days?) *Exactly, it means that trajectories start their descent at roughly 400-500hPa. Here we refer to Raveh-Rubin (2017), the process mentioned is illustrated in her Fig. 5. One can see that 24h after the start of the DI air parcels' descent, they continue their descent while simultaneously moistening. Moreover, Raveh-Rubin (2017) wrote "The originally dry air indeed gains substantial amounts of moisture, as seen by the increase of q starting at 18–24 h. This increase is possibly due to mixing with the relatively moist PBL air and/or a contribution from evaporation of sedimenting hydrometeors from overhead clouds." (p. 6669, column 1, L16).*

14. **Reviewer #2:** p. 6, section 2.2: I presume the Lagranto trajectories rely on the ERA5 subsidence velocities. Have these been compared to the EUREC4A dropsonde-circle-derived vertical velocities - is there any assessment known of the ERA5 values? *Yes, Lagranto trajectories are based on the three-dimensional ERA5 wind fields (incl. the vertical wind component). We answer this comment in the new subsection 2.3. on ERA5's representativeness and in the extension of Supplement 1, specifically Fig. S1.5 (see also item 11 of this document).*

15. **Reviewer #2**: p. 7 section 4.1: The 'Fish' cloud structure described here, similar to others, is bordered by cloud-free regions, suggesting (to me) an enhancement to the subsidence at the

mesoscale. Do the authors see cloud features like this in the ERA5 data as well? Would a spatial plot of ERA5 liquid water path provide a useful comparison to the MODIS image shown in FIg. 5? *We've looked at such plots of ERA5 CRE (exemplary time steps are shown in Fig. R5 and Fig. S1.7 of Supplement 1), total liquid and ice water content. The mesoscale cloud pattern of the Fish cloud is well reproduced in ERA5.*

[Figure]

*Fig. R5: Mesoscale cloud organisation at 14 UTC on 22 January 2020 in the domain 50-60°W, 10-20°N (a) as viewed by MODIS Terra, and as shown by total liquid water in (b) ERA5, and (c) CERES. The red circle indicates the location of BCO.*

16. **Reviewer #2:** p. 12 bottom - p. 13 top: this discussion on future work seems a better fit for the end of the manuscript. *This paragraph has been moved to Section 6. For consistency, the two sentences on L514-L516 have been moved to Section 6. For a symmetric structure of the two case studies, we also moved (and adapted) the last sentence of Subsection 5 to Section 6.*

17. **Reviewer #2:** p. 14, line 434: the references provided are all to low-level mixed-phase clouds I believe, for which the shortwave CRE will be more dominant. If the authors are able to draw in literature more pertinent to the mid-level mixed-phase cloud CRE that would be more pertinent. I provide some suggestions below but am not sure if those also evaluate the CRE. *Your suggestions have been very helpful. We have added them in the discussion.*

18. **Reviewer #2:** Figures:
    a) Fig 2: I believe this is entirely ERA5, regardless it would be useful for the figure caption to clarify. *We now indicate the data source (ERA5, satellite, radiosonde, etc.) in every Figure and Table caption in the manuscript and the Supplements.*
    b) FIg 3: I do not see the red shadings. *By the red shadings in Fig. 3, we refer to the two highlighted periods on 22 Jan and 14 Feb 2020. The two periods have a light red background in all panels, i.e. the red shading. It is marked with two red arrows in Fig. R6 (old version of Fig. 3).*

[Figure]

the red shading

**Fig. R6**: *Copy of Fig. 3 from the manuscript with the red shading indicated with red arrows.*

19. **Reviewer #2:** Other potentially useful references:
    a) Casey, S. P. F., Dessler, A. E., & Schumacher, C. (2009). Five-Year Climatology of Midtroposphere Dry Air Layers in Warm Tropical Ocean Regions as Viewed by AIRS/Aqua, Journal of Applied Meteorology and Climatology, 48(9), 1831-1842. Is their data description consistent with your ERA5 inferences? *Casey et al. (2009) analysed the frequency and source regions of dry (RH<20%) mid-tropospheric air layers (600-400 hPa; which is a bit lower in altitude than our mid-tropospheric layer) in regions with deep convection over the warm tropical oceans. Their analysis is limited to regions where the outgoing longwave radiation is below 240 W m$^{-2}$ (as an indicator for deep convection). Thus, Barbados (or the subtropics in general) is not addressed. They find that over the North Atlantic the occurrence of such dry layers is highest in boreal fall (SON) and winter (DJF) (their Fig. 3), and that the dry air source regions (according to the last saturation of 20-d backward trajectories started at 500 hPa at 00 UTC) lie over the North Atlantic (their Fig. 9). This matches our understanding of the extratropical influence. In boreal fall and summer, the influence of the extratropical storm track (as also stated by Casey et al., 2009) is larger due to the southern position of the ITCZ (compared to boreal spring and summer) and we would expect that extratropical dry intrusions frequently cause dry anomalies in the middle troposphere (however, we did not explicitly address mid-tropospheric dry layers in our study). In boreal spring (MAM) and summer (JJA), Casey et al. (2009; their Fig. 3 and 9) found that the frequency of mid-tropospheric dry layers is lower and that the trajectories (especially in JJA) followed the typical trade-wind flow (south-eastern Atlantic as source region). We cannot comment on these results, as we only investigated December and February. For all seasons, Casey et al. (2009) found North Africa as a source region for mid-tropospheric air, on p. 1837, column 2, they wrote: „Galewsky et al. (2005) and Dessler and Minschwaner (2007) identified similar trajectories in the North Atlantic and North Africa to be parcels riding up the isentropes and cooling as they mix reversibly into midlatitudes. This tropical air then condenses and loses water; when it then returns to the tropical regions, the air is drier than it was when it left.“ The excursion to midlatitudes and drying due to condensation is, in fact, exactly what we observe for the few TMDs in the quasi-climatological analysis (Fig. 18; see revised section 5.3 in the manuscript; and item 1 in this document) causing a negative anomaly in IWV$_{650-300}$ $_{hPa}$. To sum up, we cannot comment on the frequencies given by Casey et al. (2009), because we did not address dry anomalies per se, but dry anomalies associated with a specific formation pathway, which during EUREC$^4$A was observed to be linked with strong near-surface moist anomalies. However, we can comment that the formation pathways described by Casey et al. (2009) match our understanding and that we did not find anything contradicting in the formation pathways described by our ERA5 trajectories. As a further response to this comment, we now refer to his study on:*

- *L613-L615, (new text added): "Another characteristic of TMD$_{dry}$ trajectories is their excursion to comparably high latitudes (Fig. 18b), which matches one possible formation pathway of mid-tropospheric dry layers over the North Atlantic, described earlier by Casey et al. (2009)"*

a) Zuidema, P., B. Mapes, J. Lin, C. Fairall, and G. Wick, 2006: The Interaction of Clouds and Dry Air in the Eastern Tropical Pacific. J. Climate, **19**, no. 18, pp. 4531-4544. Their discussion of how the latent cooling from evaporating falling ice particles helps maintain a stability structure that can help sustain the upper-level stratiform cloud deck seems relevant to the TMD discussion on p.13. This references is very useful for the discussion on p.13. We have also added it in the introduction when the stabilizing effect of sublimation cooling at the 0°C-isotherm is introduced. In detail:
- *L124-L125 (Introduction): sentence with reference to this paper has been added*
- *L501-L502 (Sect. 5.1): sentence with reference to this paper has been added*

20. **Reviewer #2:** Relevant to the radiative impact of mid-level mixed-phase clouds:
a) Bourgeois, Q., Ekman, A. M. L. L., Igel, M. R., and Krejci, R.: Ubiquity and impact of thin mid-level clouds in the tropics, Nat. Commun., 7, 12432, https://doi.org/10.1038/ncomms12432, 2016.
b) Sassen, K. and Wang, Z.: The Clouds of the Middle Troposphere: Composition, Radiative Impact, and Global Distribution, Surv. Geophys., 33, 677–691, https://doi.org/10.1007/s10712- 011-9163-x, 2012.
c) Adebiyi, A. A., P. Zuidema, I. Chang, S. P Burton and B. Cairns, 2020: Mid-level clouds are frequent above the southeast Atlantic stratocumulus clouds. Atmos. Chem. Phys., **20**, p. 11025-11043, 10.5194/acp-20-11025-2020
*All three references are very useful for the discussion and are now referred to several times in the manuscript.*
* * *
21. **Minor changes not directly linked to a comment from a reviewer:**
- *Abstract, L21-23: Added sentence: "This is most likely due to the modulation of the CRE by above and below lying clouds and the fact that we used daily mean CRE thereby ignoring the impact of the timing of the synoptic anomaly with respect to the daily cycle."*
- *L651-L659: Slight changes in the formulation.*
- *L725-L727: added thanks to NASA for CERES data and to reviewers for the constructive feedback*
- *L730-L732: added missing funding information*
- *Correction of the variable total column ice water: We realized that in the previous version of the manuscript we used ERA5's variable "total column cloud ice water" (TCIW) instead total column ice water (TIW). Thus, we replaced TCIW with TIW, which is calculated as the sum of TCIW and vertically integrated specific snow water content. Similarly, total liquid water (TLW) has been calculated as the sum of "total column cloud liquid water" (TCLW) and the vertical integral of specific rain water content. A short paragraph addressing these variables has been added on L177-L180. For the data interpolated to the location of the BCO it doesn't matter whether TCIW or TIW is used. For TCLW and TLW, however, it does (Fig. R7).*

[Figure]

*Fig. R7: Distribution of daily mean values in the period January and February 2010-2020 of (a) total ice water (TIW) and total column cloud ice water (TCIW) and (b) total liquid water (TLW) and total column cloud liquid water (TCLW) interpolated to the location of the BCO. The boxplots show the 10-90 percentile as whiskers, the 25-75 percentile as box, the median as vertical line inside the box, and the mean value as triangles; outliers are not represented.*
* * *
**References**

Bock, O., Bosser, P., Flamant, C., Doerflinger, E., Jansen, F., Fages, R., Bony, S., and Schnitt, S. (2021). Integrated water vapour observations in the caribbean arc from a network of ground-based gnss receivers during EUREC4A. Earth Syst. Sci. Data, 13, 2407–2436. https://doi.org/10.5194/essd-13-2407-2021

Bourgeois, Q., Ekman, A. M. L., Igel, M. R., and Krejci, R. (2016). Ubiquity and impact of thin mid-level clouds in the tropics. Nat. Commun., 7, 3–8. https://doi.org/10.1038/ncomms12432

Casey, S. P. F., Dessler, A. E., & Schumacher, C. (2009). Five-year climatology of midtroposphere dry air layers in warm tropical ocean regions as viewed by AIRS/Aqua. J. Appl. Meteorol. Clim., 48, 1831–1842. https://doi.org/10.1175/2009JAMC2099.1

George, G., Stevens, B., Bony, S., Klingebiel, M., and Vogel, R. (2021). Observed impact of mesoscale vertical motion on cloudiness. J. Atmos. Sci., 78, 2413–2427. https://doi.org/10.1175/JAS-D-20-0335.1

Hersbach, H., Bell, B., Berrisford, P., Hirahara, S., Horányi, A., Muñoz-Sabater, J., Nicolas, J., Peubey, C., Radu, R., Schepers, D., Simmons, A., Soci, C., Abdalla, S., Abellan, X., Balsamo, G., Bechtold, P., Biavati, G., Bidlot, J., Bonavita, M., … Thépaut, J. N. (2020). The ERA5 global reanalysis. Q. J. Roy. Meteor. Soc., 146, 1999–2049. https://doi.org/10.1002/qj.3803

Holton, J. R., and Hakim, G. J. (2013). Tropical dynamics. In J. R. Holton and G. J. Hakim (Eds.), Introduction to dynamic meterology (5th ed., pp. 377–411). Academic Press.

NASA/LARC/SD/ASDC. (2017). CERES and GEO-Enhanced TOA, Within-Atmosphere and Surface Fluxes, Clouds and Aerosols 1-Hourly Terra-Aqua Edition4A [Data set]. NASA Langley Atmospheric Science Data Center DAAC. https://doi.org/10.5067/TERRA+AQUA/CERES/SYN1DEG-1HOUR_L3.004A

Raveh-Rubin, S. (2017). Dry intrusions: Lagrangian climatology and dynamical impact on the planetary boundary layer. J. Climate, 30, 6661–6682. https://doi.org/10.1175/JCLI-D-16-0782.1

Romps, D. M. (2017). Exact expression for the lifting condensation level. J. Atmos. Sci., 74, 3891–3900. https://doi.org/10.1175/JAS-D-17-0102.1

Salathé, E. P., and Hartmann, D. L. (1997). A trajectory analysis of tropical upper-tropospheric moisture and convection. J. Climate, 10, 2533–2547. https://doi.org/10.1175/1520-0442(1997)010<2533:ATAOTU>2.0.CO;2

Sassen, K., and Wang, Z. (2012). The clouds of the middle troposphere: composition, radiative impact, and global distribution. Surv. Geophys., 33, 677–691. https://doi.org/10.1007/s10712-011-9163-x

Schemm, S., Rudeva, I., and Simmonds, I. (2015). Extratropical fronts in the lower troposphere-global perspectives obtained from two automated methods. Q. J. Roy. Meteor. Soc., 141, 1686–1698. https://doi.org/10.1002/qj.2471

Schulz, H., Eastman, R., and Stevens, B. (2021). Characterization and evolution of organized shallow convection in the trades. Earth Space Sci. Open Archive [Preprint]. https://doi.org/10.1002/essoar.10505836.1

Zuidema, P., Mapes, B., Lin, J., Fairall, C., and Wick, G. (2006). The interaction of clouds and dry air in the eastern tropical Pacific. J. Climate, 19, 4531–4544. https://doi.org/10.1175/JCLI3836.1